# Chidamide plus Tyrosine Kinase Inhibitor Remodel the Tumor Immune Microenvironment and Reduce Tumor Progression When Combined with Immune Checkpoint Inhibitor in Naïve and Anti-PD-1 Resistant CT26-Bearing Mice

**DOI:** 10.3390/ijms231810677

**Published:** 2022-09-14

**Authors:** Jia-Shiong Chen, Yi-Chien Hsieh, Cheng-Han Chou, Yi-Hong Wu, Mu-Hsuan Yang, Sz-Hao Chu, Ye-Su Chao, Chia-Nan Chen

**Affiliations:** 1New Drug Research and Development Center, Great Novel Therapeutics Biotech & Medicals Corporation (GNTbm), Taipei 100, Taiwan; 2Wesing Breast Hospital, Kaohsiung 800, Taiwan; 3Department of Biology, Great Novel Therapeutics Biotech & Medicals Corporation (GNTbm), Taipei 100, Taiwan; 4Department of Chemistry, Great Novel Therapeutics Biotech & Medicals Corporation (GNTbm), Taipei 100, Taiwan

**Keywords:** tumor microenvironment, VEGF receptor tyrosine kinase inhibitors, class l histone deacetylase inhibitor, colon cancer, immune checkpoint inhibitors

## Abstract

Combined inhibition of vascular endothelial growth factor receptor (VEGFR) and the programmed cell death protein 1 (PD-1) pathways has shown efficacy in multiple cancers; however, the clinical outcomes show limited benefits and the unmet clinical needs still remain and require improvement in efficacy. Using murine colon carcinoma (CT26) allograft models, we examined the efficacy and elucidated novel tumor microenvironment (TME) remodeling mechanisms underlying the combination of chidamide (a benzamide-based class l histone deacetylase inhibitor; brand name in Taiwan, Kepida^®^) with VEGF receptor tyrosine kinase inhibitor (TKIs; cabozantinib/regorafenib, etc.) and immune checkpoint inhibitors (ICIs; anti-PD-1/anti-PD-L1/anti-CTLA-4 antibodies). The TME was assessed using flow cytometry and RNA-sequencing to determine the novel mechanisms and their correlation with therapeutic effects in mice with significant treatment response. Compared with ICI alone or cabozantinib/regorafenib + ICI, combination of chidamide + cabozantinib/regorafenib + ICI increased the tumor response and survival benefits. In particular, treatment of CT26-bearing mice with chidamide + regorafenib + anti-PD-1 antibody showed a better objective response rate (ORR) and overall survival (OS). Similar results were observed in anti-PD-1 treatment-resistant mice. After treatment with this optimal combination, in the TME, RNA-sequencing revealed that downregulated mRNAs were correlated with leukocyte migration, cell chemotaxis, and macrophage gene sets, and flow cytometry analysis showed that the cell numbers of myeloid-derived polymorphonuclear suppressor cells and tumor-associated macrophages were decreased. Accordingly, chidamide + regorafenib + anti-PD-1 antibody combination therapy could trigger a novel TME remodeling mechanism by attenuating immunosuppressive cells, and restoring T-cell activation to enhance ORR and OS. Our studies also showed that the addition of Chidamide to the regorafenib + anti-PD-1 Ab combination could induce a durable tumor-specific response by attenuating immune suppression in the TME. In addition, this result suggests that TME remodeling, mediated by epigenetic immunomodulator combined with TKI and ICI, would be more advantageous for achieving a high objective response rate, when compared to TKI plus ICI or ICI alone, and maintaining long-lasting antitumor activity.

## 1. Introduction

Previous reports have established that several types of cancer immunotherapy, including chimeric antigen receptor T-cells (CAR-T) and immune checkpoint inhibitor (ICI) monoclonal antibodies, can reinvigorate antitumor T-cells and dynamically modulate anticancer immune responses [1,2]. Ipilimumab, the first antibody to block an immune checkpoint (anti-CTLA-4 antibody), was approved and rapidly followed by targeted PD-1 and PD-L1 inhibition in T-cells [3]. To date, these antibodies have received market approval for treating 23 types of cancer and have become some of the most widely prescribed anticancer therapies [4]. Patients with metastatic melanoma can achieve durable and complete remission after discontinuing pembrolizumab, and the low incidence of relapse after a median follow-up of approximately two years post-discontinuation provides hope for a cure in some patients [5]. Over the past several years, ICIs have revolutionized the treatment of advanced melanoma, non-small-cell lung carcinoma, and renal cell carcinoma (RCC), and have ensured significant improvement in OS, when compared with standard chemotherapy [6]. However, one major persistent challenge is that some tumors that initially respond to ICI treatment ultimately develop acquired resistance, subsequently resulting in tumor progression [7]. In addition, although these new ICI therapies have improved outcomes in numerous patients, a significant proportion of patients still experience a lack of response, known as primary resistance [8]. Furthermore, some patients may fail to exhibit expected outcomes, exhibiting rapid cancer progression, called hyperprogressive disease (HPD), and reportedly observed in patients with advanced gastric cancer treated with anti-PD-1 monoclonal antibody [9]. Up to date, according to trial information listed in ClinicalTrials gov, there are more than 300 clinical trials related to immune checkpoint inhibitors in combination with other therapies in development stages of Phase 2 or 3 for treatment of cancer patients who are resistant to anti-PD-1 therapy or for improvement of the response rate. The combination strategies of ICI include combinations such as ICI plus chemotherapy, targeted drugs, radiotherapy, etc., or, even, combinations with a different ICI.

Resistance to cancer immunotherapy can be attributed to the dynamic components and composition of the TME. Tumor cells can create an immunosuppressive phenotype that promotes vitality and tumor expansion. It has been shown that tumor immunogenicity can be influenced by tumor mutational burden (TMB), T-cell priming processes, and precise functioning of antigen-presenting cells (APC)/dendritic cells (DC); any aberration in these factors could enable tumor cell immune evasion, especially in melanomas [10]. Primary tumor resistance toward ICIs can be attributed to low TMB, resulting in a lack of tumor-associated antigen presentation, reduced T-cell priming, and weak tumor infiltration [11]. In addition, tumor cells can impair DC maturation and recruit immunosuppressive cells (myeloid-derived suppressor cells [MDSCs] and regulatory T-cells [Tregs]) by maintaining interleukin (IL)-6, IL-10, and vascular endothelial growth factor (VEGF)-A levels in the TME, consequently attenuating the T-cell response against tumors [12,13,14]. However, acquired tumor resistance to ICIs is more complex than primary tumor resistance. A previous study has suggested that the loss of immunogenic neoantigens, due to mutation of the histocompatibility complex I (MHC I) molecule, and overexpression of interferon-gamma (IFN-γ)-induced T-cell-silencing ligands, such as programmed death-ligand 1 (PD-L1), lymphocyte activation gene-3 (LAG-3), and T-cell immunoglobulin and mucin domain-3 (TIM-3), after ICI treatment, can promote inhibitory signals in effector T-cells, and result in exhausted tumor-infiltrating lymphocytes (TILs) losing their ability to attack tumor cells [15,16]. Tumors associated with HPD showed augmented activation and expansion of tumor-infiltrating PD-1+ Treg cells after PD-1 blockade treatment, as well as TIM3 overexpression in tumor-infiltrating CD8+ T-cells, resulting in T-cell exhaustion. In addition, mice deficient in PD-1 signaling showed rapid proliferation of FoxP3+ Treg cells, thus implicating immunosuppression following PD-1 blockade therapy [9,17].

Anti-angiogenesis drug-mediated vascular normalization can result in indirect physical effects, which lead to reduced hypoxia, increased immune-cell infiltration, and decreased expression of PD-1 on CD8+ T-cells in the TME [18,19]. This concept provides a novel rationale for the immunomodulatory role of anti-angiogenesis targeting VEGF-A/VEGF receptor (VEGFR) for overcoming ICI primary resistance. Reportedly, combining an anti-angiogenesis drug with an anti-PD-L1 antibody can induce strong and synergistic antitumor responses by limiting T-cell exhaustion in small cell lung cancer [20]. In addition, previous studies have demonstrated that apatinib (a well-known TKI) combined with an anti-PD-1 antibody could increase CD8+ TIL cells and restore function in exhausted CD8+ T-cells in anti-PD-1 antibody-resistant cancers [21]. Thus, anti-angiogenesis therapy could improve the efficacy of ICIs in VEGF-rich tumors, such as liver and colon cancers. Several clinically developed TKIs, including axitinib, sunitinib, lenvatinib, cabozantinib, and regorafenib, have been approved by the US Food and Drug Administration (FDA). However, a previous study has shown that long-term treatment with anti-VEGFR2 antibody can induce an immunosuppressive TME [22] and sunitinib leads to higher infiltration of Treg cells, and upregulated PD-L1 expression, and is related to disappointing outcomes in RCC patients [23]. Moreover, under hypoxic conditions, PD-L1 expression is rapidly upregulated in immunosuppressive cells in a hypoxia-inducible factor 1α (HIF-1α)-dependent manner [24]. Hypoxia-induced metabolic changes, particularly at a relatively low-pH microenvironment, could reduce the lifespan of CD8+ memory T-cells, impair natural killer (NK) cell activation, and MDSC recruitment [25]. Intratumor hypoxia and the related infiltration of immunosuppressive cells seem to be largely responsible for angiogenetic relapse and drug resistance. Therefore, a third combination therapeutic strategy, focused on the combined agents that target the immunosuppressive phenotype in TME, would potentially overcome cancer immunotherapy-induced resistance.

A previous study has reported that HIF-1α inhibition provides adjuvant immune activity, thereby improving the efficacy of tumor antigen-based DC vaccines by augmenting the proliferation and function of cytotoxic T lymphocytes and increasing IFN-γ production in a breast cancer model [26]. In addition, pan-histone deacetylase (HDAC) inhibitor (HDACi; LAQ824) could promote the polyubiquitination of HIF-1α via yet unknown mechanisms, thereby inhibiting the function of HIF-α [27]. Inhibition of HIF-1 is beneficial for regulating the TME and conducive to antitumor activity. Moreover, HDAC1 expression positively correlates with HIF-1α expression, and these changes are associated with poor prognosis in patients with advanced cancer [28]. These results suggest that class I HDACis may afford protection against hypoxia-induced HIF-1α accumulation, remodel the TME in combination with anti-angiogenesis drugs, and overcome resistance by modulating the immunosuppressive phenotype in the TME. Accordingly, our hypothesis and data support the view that triple combination of ICIs+ VEGFR-TKIs + class I HDACis (chidamide) affords potential antitumor activity in naïve and anti-PD-1 antibody treatment-resistant CT26-bearing mice.

## 2. Results

### 2.1. Anticancer Effect of VEGFR-TKIs Combined with Anti-PD-1 Ab in CT26-Bearing Mice

Herein, we initially studied the anticancer activity of small-molecule TKI monotherapy and evaluated the potential of TKIs plus chidamide combination therapy to enhance the objective response rate in CT26-bearing mice models. As shown in Appendix AA–C, treatment with lenvatinib, cabozantinib, regorafenib, or axitinib alone, only partially reduced tumor growth. For the above 4 drugs, cabozantinib or regorafenib when combined with chidamide markedly inhibited tumor growth when compared with lenvatinib/axitinib combined with chidamide, as shown in Appendix AC. Mice treated with the different regimens showed no significant loss in body weight, as shown in Appendix AD. Individual tumor sizes (fold-change) and ORR, as shown in Appendix AE, indicated that the monotherapy groups achieved 25–38% ORR; cabozantinib combined with chidamide therapy achieved 88% ORR, while regorafenib combined with chidamide also achieved 88% ORR. As shown in Appendix AF, cabozantinib/regorafenib combined with chidamide exerted a good survival rate. Therefore, cabozantinib/regorafenib combined with chidamide treatment achieved great antitumor effect and may exhibit even greater antitumor activity in combination with anti-PD-1 antibody in immunotherapy application.

Our previous study suggested that chidamide and celecoxib are potent TME modulators that improve antitumor activity when combined with anti-PD-1 antibody immunotherapy [29]. PGE2 was shown to activate several key immuno-suppressive cells present in the TME, such as Treg, M-MDSC, and TAM. Therefore, celecoxib, inhibiting PGE2 synthesis, may be used as a co-regulator of the tumor microenvironment. While evaluating the anticancer effect of cabozantinib plus chidamide in combination with anti-PD-1 antibody, we also evaluated whether anti-PD-1 antibody combined with cabozantinib plus celecoxib could achieve the same effect. As shown in Appendix AA–C, the anti-PD-1 antibody combined with cabozantinib plus celecoxib or chidamide-k30 regimen achieved excellent tumor growth inhibition when compared with the anti-PD-1 antibody treatment alone. Celecoxib is a selective cyclooxygenase (COX)-2 inhibitor and chidamide is a potent benzamide-based HDACi that selectively inhibits HDACs 1, 2, 3, and 10. We observed no significant loss in body weight in mice treated with the different regimens, as shown in Appendix AD. The individual tumor sizes (fold-change) and ORR, as shown in Appendix AE, indicated that treatment with anti-PD-1 antibody group achieved 25% ORR, anti-PD-1 antibody combined with cabozantinib plus celecoxib achieved 57% ORR, and anti-PD-1 antibody (2.5 mg/kg) combined with cabozantinib plus chidamide-k30 achieved 100% ORR. These findings demonstrated that the mice treated with anti-PD-1 antibody combined with cabozantinib plus chidamide-k30 achieved better ORR than those treated with anti-PD-1 antibody combined with cabozantinib plus celecoxib. This is the first study to report that TKI plus HDACi combined with ICI significantly enhanced the response rate. As shown in Appendix AF, anti-PD-1 antibody combined with cabozantinib plus chidamide-k30 markedly prolonged survival when compared with the other groups, achieving a 100% survival rate by day 60.

### 2.2. Anticancer Activity of Anti-PD-1 Antibody Combined with Cabozantinib or Regorafenib Plus Chidamide-k30 in CT26-Bearing Mice

As shown in Appendix A, our results revealed that chidamide-k30 combined with anti-PD-1 antibody plus cabozantinib potently inhibited tumor growth. Next, we examined anti-PD-1 antibody therapy combined with regorafenib (like cabozantinib, a potent TKI and second-line treatment for advanced liver cancer approved by the FDA in 2017; third-line treatment for advanced colon cancer approved by the FDA in 2012) plus chidamide-k30 regimens in CT26 tumor-bearing mice model. As shown in Figure 1A–C, anti-PD-1 antibody combined with cabozantinib plus chidamide-k30 regimen more effectively inhibited tumor growth than anti-PD-1 antibody combined with cabozantinib regimens, as expected. This result suggested that, in combination with cabozantinib, chidamide may be potently effective, not inferior to anti-PD-1 antibody, as shown in Figure 1C. Similar results were also observed in the regorafenib treatment groups; anti-PD-1 antibody combined with regorafenib plus chidamide-k30 was more effective in inhibiting tumor growth than anti-PD-1 antibody combined with regorafenib. As shown in Figure 1D, treatment with regorafenib plus chidamide-k30 combined with anti-PD-1 antibody initially reduced body weight, which was eventually recovered after continuous treatment. Individual tumor sizes (fold-change) and ORR, as shown in Figure 1E, indicated that anti-PD-1 antibody combined with cabozantinib plus chidamide-k30 achieved 50% ORR. The group treated with anti-PD-1 antibody combined with regorafenib plus chidamide-k30 achieved 43% ORR. The survival rate was evaluated in the anti-PD-1 antibody combined with cabozantinib/regorafenib plus chidamide-k30 group. As shown in Figure 1F, anti-PD-1 antibody combined with regorafenib plus chidamide-k30 was more potent in prolonging survival than anti-PD-1 antibody combined with cabozantinib plus chidamide-k30. This result suggested that chidamide is a crucial component for improving the anti-PD-1 antibody combined with the cabozantinib/regorafenib regimen to significantly enhance the ORR and survival rate in CT26 tumor-bearing mice. Several clinical trials have examined anti-PD-1/anti-PD-L1 antibody therapy combined with the cabozantinib/regorafenib regimens to boost ORR and OS in advanced cancers, such as RCC, hepatocellular carcinoma (HCC), and metastatic colorectal cancer (mCC). Interestingly, the anti-PD-1 antibody combined with regorafenib plus chidamide-k30 regimen exhibited 43% ORR, but the OS rate approximated 86%. We also observed that, although treatment with this drug combination was discontinued, the tumor persistently shrunk (as shown in Table 1). It was hypothetically suggested that this regimen could robustly modulate tumor immunologic activity. This effect is the most desirable result of cancer immunotherapy, activating the immune system with drugs that produce long-lasting immune memories that continue attacking tumor cells. To test the proposed mechanism, the observation of recurrence and rechallenge studies were performed, as outlined in Figure 1A. Based on the preliminary results of the recurrence rate, as shown in Table 1, anti-PD-1 antibody combined with regorafenib plus chidamide-k30 regimen seemed to be more effective in preventing relapse than anti-PD-1 antibody combined with cabozantinib plus chidamide-k30 regimen. Furthermore, in the rechallenge study, we found that the mice with CR or PR response after treatment with anti-PD-1 antibody combined with cabozantinib or regorafenib plus chidamide-k30 showed resistance to CT26 rechallenge. Based on the results of these studies, it could be suggested that regimens comprising anti-PD-1 antibody combined with multi-kinase inhibitor, such as regorafenib or cabozantinib, plus chidamide-k30, activated a specific immune memory, thereby exerting strong anticancer activity. These findings indicated that modulation of the TME could improve the tumor response rate, avoid recurrence, and possibly induce immune memory.

### 2.3. Chidamide Is a Key Component in Triplet Combination Regimens for Significantly Regulating Immune Cell Population and Gene Expression in the TME of CT26 Tumor-Bearing Mice

We next investigated whether treatment with a double regimen of anti-PD-1 antibody combined with cabozantinib/regorafenib, or a triple regimen comprising anti-PD-1 antibody combined with cabozantinib/regorafenib plus chidamide-k30, affected immune cells, including myeloid-cell and T-cell populations, in tumors. Accordingly, we examined tumor samples isolated on day 20 after 9-day treatment and assessed immune cells using flow cytometry (FACS) in Appendix A. The anticancer activities for all treatment regimens were assessed on day 20 before tumors were excised, and are shown in Figure 2B. As shown in Figure 2C–F, populations of CD3+, CD4+ T-cells, and Tregs in tumors were significantly altered following treatment with anti-PD-1 antibody combined with cabozantinib or regorafenib, with or without chidamide-k30; however, anti-PD-1 antibody+ regorafenib+ chidamide-k30 treatment did not significantly decrease Tregs, while markedly increasing CD8+ T-cells. These results suggested that anti-PD-1 antibody combined with cabozantinib or regorafenib effectively reduced Tregs, which was favorable for activating tumor immunity in response to tumor inhibition. In addition, only the PD-1 antibody + regorafenib + chidamide-k30 regimen significantly increased CD8+ T-cell infiltration; this was deemed more conducive for inhibiting tumor growth. In the MDSC population, compared with other regimens (anti-IgG and anti-PD-1 groups), anti-PD-1 antibody combined with regorafenib plus chidamide-k30 treatment significantly decreased PMN-MDSCs and TAMs in tumors, as shown in Figure 2G,H,J. Accordingly, the results of the anti-PD-1 antibody + regorafenib + chidamide-k30 triple combination were likely attributed to the altering of tumor-infiltrating immune cells, increasing CD8+ T-cells and decreasing immune-suppressive PMN-MDSCs and TAMs in tumors. However, the level of tumor M-MDSCs exhibited no correlation with anticancer activity in double or triple combination regimens, when comparing the results of Figure 2B,I. Taken together, in terms of TME regulation, tumor growth could be inhibited as long as a relative regulatory advantage could be achieved, rather than an absolute advantage, achieved.

Gene expression analysis of CT26 tumors revealed that multiple immune-related pathways were induced following treatment with anti-PD-1 antibody combined with cabozantinib/regorafenib, with or without chidamide. The consecutive treatment schedule and tumor size of each treatment group are shown in Appendix AA,B. To clarify molecular mechanisms underlying the double and triple regimen combination in cancer immunotherapy, we performed RNA-sequencing (RNA-seq) on CT26 tumors measured and excised on day 20 after 9-day treatment (Appendix AA,B) to identify regulated genes. On analyzing our RNA-seq data, we identified mRNA transcript levels of genes upregulated and downregulated by the above treatment regimens (Figure 3A). The numbers of genes upregulated and downregulated after each regimen treatment are labeled on the top-right and top-left corners of Figure 3A, respectively. As shown in Figure 3B, GO analysis was performed on upregulated and downregulated genes, given that the minimal fold-change was 2 and *p*-adjusted was <0.05. Based on DEG analysis, we also performed GSEA enriched analysis, showing that the pathways possibly involved were those including T-cell-mediated cytotoxicity, angiogenesis, chemokine activity, cytokine activity, leukocyte migration, and cell chemotaxis. Leukocyte migration is a major target of triple combination treatment regimens anti-PD-1 antibody combined with cabozantinib/regorafenib plus chidamide-k30. Therefore, we focused on analyzing chemokine and cytokine activity/gene expression. GSEA of DEGs highlighted enrichment of pathways in anti-PD-1 antibody-treated tumors, with upregulated T-cell-mediated cytotoxicity response gene signature; however, with downregulated angiogenesis response gene signature (Figure 3C; *p* < 0.05). In PD-1 antibody + cabozantinib treated tumors, we noted upregulated enrichment of the response to T-cell-mediated cytotoxicity, chemokine activation, cytokine activity gene signature, with downregulated angiogenesis, leukocyte migration, and cell chemotaxis response gene signature (Figure 3D; *p* < 0.05). In PD-1 antibody + cabozantinib + chidamide-k30 treated tumors, we noted upregulated enrichment of the response to chemokine activation, cytokine activity, leukocyte migration, cell chemotaxis, and T-cell-mediated cytotoxicity response gene signature, with downregulated angiogenesis response gene signature (Figure 3E; *p* < 0.05). In PD-1 antibody + regorafenib treated tumors, upregulated enrichment of the response to chemokine activation and cytokine activity was observed, with downregulated leukocyte migration cell chemotaxis and angiogenesis response gene signature (Figure 3F; *p* < 0.05). In PD-1 antibody + regorafenib + chidamide-k30 treated tumors, we observed the upregulated enrichment of the response to chemokine activation, cytokine activity, and T-cell-mediated cytotoxicity, along with downregulated leukocyte migration, cell chemotaxis and angiogenesis response gene signature (Figure 3G; *p* < 0.05). Compared with anti-IgG antibody therapy, all treatments significantly downregulated pathways involved in angiogenesis (Figure 3C–G).

Compared with the anti-PD-1 antibody alone, the triple regimen comprising anti-PD-1 antibody + cabozantinib/regorafenib + chidamide-k30 more significantly showed their effect on the upregulation of chemokine, cytokine activity-related and T-cell-mediated cytotoxicity gene expression. Reportedly, the IFN-γ response is critical for activating TILs [30]. Therefore, GSEA was performed using published gene sets for the immune cell signature. The IFN pathway and T-cell gene signature was upregulated in anti-PD-1 antibody-treated tumors (Appendix AC; *p* < 0.001). In anti-PD-1 antibody + cabozantinib treated tumors, we observed a trend of upregulated IFN pathway gene signature (Appendix AD; *p* < 0.001). In addition, PD-1 + cabozantinib + chidamide-k30 treated tumors showed significantly upregulated IFN pathway, macrophage gene, neutrophil gene, and T-cell gene signatures but downregulated enrichment of the monocyte gene signature (Figure 4E; *p* < 0.05). In PD-1 + regorafenib treated tumors, enrichment of the macrophage, monocyte, and T-cell gene signature was significantly downregulated (Appendix AF; *p* < 0.05). In PD-1 + regorafenib + chidamide-k30 treated tumors, we detected upregulated enrichment of IFN pathway gene signature and downregulated enrichment of the macrophage and T-cell gene signatures (Appendix AG; *p* < 0.001). These results demonstrated that intratumor heterogeneity of gene expression elicited responses to anti-PD-1 antibody + regorafenib/cabozantinib + chidamide-k30 therapy in CT26-bearing mice models. Anti-PD-1 antibody + cabozantinib did not significantly downregulate or upregulate macrophage gene and monocyte gene signatures; however, the downregulation of monocyte gene signature and upregulation of macrophage gene, neutrophil gene, and T-cell gene signatures were enhanced by further combination with chidamide. Anti-PD-1 antibody + regorafenib significantly downregulated macrophage, monocyte, and T-cell gene sets, but did not induce IFN response gene set. However further addition of chidamide to the anti-PD-1 antibody + regorafenib combination attenuated this downregulation effect on monocytes and induced IFN-γ response gene set. Collectively, our data suggested that anti-PD-1 antibody combined with regorafenib plus chidamide modulated immune cell migration, angiogenesis, and cytokine, or chemokine, gene expression activity in the TME to enhance the tumor response rate in CT26 tumor-bearing mice.

### 2.4. Anticancer Activity of Several ICIs Combined with Cabozantinib/Regorafenib Plus Chidamide in CT26-Bearing Mice

Next, we evaluated the activities of tumor growth inhibition following treatment with different ICIs, such as anti-PD-1/anti-PD-L1/anti-CTLA-4 antibodies, combined with regorafenib/cabozantinib plus chidamide-k30 regimens in CT26 tumor-bearing mice. As shown in Figure 4A–C, the anti-PD-L1 antibody combined with regorafenib plus chidamide-k30 regimen more effectively inhibited tumor growth than the anti-CTLA-4 or anti-PD-1 antibody combination regimens. However, the anti-CTLA4 antibody combined with the cabozantinib plus chidamide-k30 regimen was more effective for inhibiting tumor growth than the anti-PD-L1 or anti-PD-1 antibody combination regimens. As shown in Figure 4D, most triple combination regimens partially reduced body weight on day 20, except for anti-PD-1+ cabozantinib plus chidamide-k30; however, body weight gradually recovered after the last treatment administration on day 26. Individual tumor sizes (fold-change) and ORR, as shown in Figure 4E, indicated that anti-PD-L1 antibody combined with regorafenib plus chidamide-k30 achieved 89% ORR, whereas treatment with anti-CTLA-4 antibody combined with cabozantinib plus chidamide-k30 achieved 90% ORR; these two regimens were superior to the other regimens. The anti-PD-1 antibody combined with regorafenib/cabozantinib plus chidamide-k30 regimen showed relatively poorer ORR than the other regimens. As shown in Figure 4F, different ICI combinations with cabozantinib/regorafenib plus chidamide-k30 exhibited distinct OS rates. The medians of all triple combination regimens exceeded 61 days. Interestingly, two combinations (anti-PD-1 antibody combined with regorafenib plus chidamide-k30 and anti-PD-1 antibody combined with cabozantinib plus chidamide-k30) showed good OS, even with a moderate ORR (~30 to 40%); this finding indicated that these two triple regimens might potently activate the immune system to inhibit cancer cell growth and avoid relapse, as shown in Table 2. Surprisingly, after discontinuing drug administration, the tumor in CT26 tumor-bearing mice continued shrinking; therefore, a second ORR assessment was performed 10 days after the last drug administration. Treatment with anti-PD-1 antibody combined with regorafenib plus chidamide-k30 significantly enhanced ORR from 30% to 60% in the second ORR assessment, indicating that the immune system was potentially induced for the direct or indirect activation of CD8+ T-cells to attack tumor cells. A similar phenomenon was also observed following treatment with anti-CTLA-4 antibody combined with regorafenib plus chidamide-k30, presenting an enhanced ORR (increased from 60% to 80%) and a greater number of mice achieving CR. These data suggested that anti-PD-1/anti-PD-L1/anti-CTLA-4 antibody combined with regorafenib/cabozantinib plus chidamide-k30 possessed markedly potent activity in terms of activating the immune system. Hence, it should be noted that ICI combined with VEGFR-TKIs plus chidamide could continue alleviating tumor growth in mice who initially achieved SD or PR (first assessment). That is, despite drug discontinuation, these triple combination regimens could improve SD to PR or PR to CR in CT26-bearing mice. Finally, immunity was further evaluated using a rechallenge experiment (as shown in the far-right column of Table 2). Overall, the rechallenge experiment revealed that almost all triple combination regimens could significantly activate immunity, thereby effectively suppressing the rechallenge-induced proliferation of CT26 cancer cells.

### 2.5. Overcoming First-Line Anti-PD-1 Antibody Treatment-Induced Drug Resistance Using Cabozantinib/Regorafenib plus Chidamide Combined with Anti-CTLA-4 Antibody in CT26-Bearing Mice

In the present study, treatment with anti-CTLA-4 antibody combined with regorafenib or cabozantinib plus chidamide-k30 exhibited potent antitumor activity. We next examined whether this therapeutic benefit could also be observed in anti-PD-1 therapy resistant mice. Mice were treated with second-line therapy to mimic the treatment strategy employed for first-line drug-induced resistance known to occur in humans. It is well-established that numerous patients with cancer receiving first-line anti-PD-1 antibody therapy develop resistance, including primary resistance, acquired resistance, or HPD. To evaluate the effectiveness of different second-line treatments after development of drug resistance to first-line anti-PD-1 antibody, a platform with a suitable treatment schedule was designed, as outlined in Figure 5A. As shown in Figure 5A,B, first, 119 mice were treated with anti-PD-1 antibody as the first-line treatment, and 10 mice were treated with anti-IgG antibody as a negative control. Overall, 17 of the 119 mice achieved response to first-line anti-PD-1 antibody treatment, achieving a 14.3% ORR (17/119). Among the 119 mice, 102 showed primary resistance (non-response) to first-line anti-PD-1 antibody treatment, with an occurrence rate of 76.5% (91/119), as shown in Table 3. Regarding the HPD mice (with an incidence of approximately 9.2% [11/119]), defined accordingly, based on tumor volumes >600 mm^3^, the average tumor volume was 669 mm^3^, as shown in Figure 5A. These results suggested that primary resistance and HPD to treatment with the first-line anti-PD-1 antibody remains a challenging issue in cancer immunotherapy. Mice with primary resistance and HPD to anti-PD-1 antibody therapy were randomized to nine different treatment groups (*n* = 5–11 mice/group), as indicated. As our unpublished data demonstrated, CC-02 (chidamide-HCL + celecoxib) combined with an anti-CTLA4 antibody was shown to elicit a significant anticancer activity for second-line treatment; and the chloride salt formulation of chidamide significantly improved aqueous solubility compared with the k30 coated formulation. As shown in Figure 5C–E, both regimens, i.e., anti-CTLA-4 antibody combined with regorafenib plus chidamide-k30 and anti-CTLA-4 antibody combined with cabozantinib plus chidamide-k30, were more potent in inhibiting tumor growth than the positive controls anti-CTLA-4 antibody combined with CC-02 and anti-CTLA-4 antibody alone. These results suggested that anti-CTLA-4 antibody combined with regorafenib/cabozantinib plus chidamide-k30 potently regulated TME activity to overcome primary resistance to first-line anti-PD-1 antibody therapy. As shown in Figure 5F, no significant body weight loss was detected in all treatment groups, except the anti-CTLA-4 antibody + cabozantinib plus chidamide-k30 group, in which body weight was initially partially reduced and then gradually recovered. The HPD mice were treated with anti-CTLA-4 antibody combined with cabozantinib plus chidamide-HCl salt for 16 days, and large tumors surprisingly showed growth inhibition, while some tumors were controlled by persistent shrinkage under this treatment regimen, as shown in Figure 5D. These results suggested that the anti-CTLA-4 antibody combined with chidamide-HCl salt plus cabozantinib regimen could potently rescue HPD (none of the mice achieved PD). Individual results for each drug-resistant mouse are shown in Figure 5G. Treatment with CC-02 achieved 37.5% ORR. Treatment with anti-CTLA-4 antibody combined with CC-02 as positive control showed 37.5% ORR. However, treatment with anti-CTLA-4 antibody combined with regorafenib plus chidamide-k30 achieved 62.5% ORR. Treatment with anti-CTLA-4 antibody combined with cabozantinib plus chidamide-k30 achieved 57.1% ORR. In addition, treatment of HPD mice with anti-CTLA-4 antibody combined with cabozantinib plus chidamide-HCl salt showed 18.1% ORR. Although the tumor response rates of HPD mice were lower than those of other combination groups, most tumor-bearing mice showed inhibited tumor growth after treatment with this second-line regimen. This result was intriguing because HPD tumors grow more rapidly and are difficult to effectively reduce and inhibit. Furthermore, we evaluated the survival rate of mice with primary resistance or HPD to anti-PD-1 antibody treatment. As shown in Figure 5H,I, after discontinuing second-line therapy, treatment with anti-CTLA-4 antibody combined with regorafenib plus chidamide-k30 achieved an OS of 87.5%, while anti-CTLA-4 antibody combined with cabozantinib plus chidamide-k30 group achieved an OS of 71.4%, thus indicating possible immune system activation to inhibit tumor cell growth and avoid relapse, as shown in Table 4. Regarding HPD mice, treatment with anti-CTLA-4 antibody combined with cabozantinib plus chidamide-HCl salt achieved a good OS rate (45.4%). This result was noteworthy because HPD mice only achieved 18.1% ORR but achieved an OS rate of 45.4% due to significant suppression of tumor growth (Table 4). Finally, immunity was further examined using a rechallenge experiment (as shown in the far-right column of Table 4) and revealed that almost all regimens were significantly active in stimulating immunity, thereby effectively inhibiting the rechallenge-induced proliferation of CT26 cancer cells.

### 2.6. IntraTumor Gene Expression in First-Line Anti-PD-1 Antibody-Resistant CT-26 Tumor-Bearing Mice after Second-Line Therapy with Chidamide-k30 Combined with Cabozantinib/Regorafenib Plus Anti-CTLA-4 Antibody

Appendix AA presents the consecutive treatment schedule for the second-line treatment group. To clarify TME remodeling induced by double and triple combination regimens in cancer immunotherapy, we performed RNA-seq analysis on CT26 tumors excised on day 28 after 12-day treatment. Using RNA-seq data, we identified upregulated and downregulated mRNA transcript levels in each treatment group compared with IgG (Figure 6A). GO analysis was performed on upregulated and downregulated genes, given the minimal fold-change was 2 and *p*-adjusted was <0.05, which were associated with the responses to IFN-γ, cellular response to IFN-γ, leukocyte migration, cell chemotaxis, and adaptive immune response, etc. (Figure 6B). GSEA of DEGs highlighted enrichment for pathways in second-line anti-CTLA-4 antibody-treated tumors, presenting upregulation of cell chemotaxis and leukocyte migration gene signatures (Figure 6C; *p* < 0.05). In second-line CC-02 treated tumors, the responses to IFN-γ, cellular response to IFN-γ, adaptive immune response, leukocyte migration and cell chemotaxis were upregulated, whereas angiogenesis gene signature was downregulated (Figure 6D; *p* < 0.05). The second-line anti-CTLA-4 + CC-02 treated tumors showed upregulation of the response to IFN-γ, cellular response to IFN-γ, adaptive immune response gene signature, leukocyte migration and cell chemotaxis (Figure 6E; *p* < 0.05). The second-line triple combinations anti-CTLA-4 + cabozantinib + chidamide-k30 tumor treatment upregulated the response to IFN-γ, cellular response to IFN-γ, adaptive immune response gene signatures, cell chemotaxis and leukocyte migration gene signatures; however, it downregulated angiogenesis gene signature (Figure 6F; *p* < 0.05). In second-line anti-CTLA-4 + regorafenib + CD-k30 treated tumors, we noted upregulated response to IFN-γ, cellular response to IFN-γ, leukocyte migration and cell chemotaxis, and downregulated adaptive immune response and angiogenesis gene signature (Figure 6G; *p* < 0.05). Compared with anti-IgG antibody, except for anti-CTLA4 antibody and anti-CTLA-4 + CC-02 triple combination, CC-02 and triple combinations anti-CTLA-4 + regorafenib/ cabozantinib + CD-k30 significantly downregulated pathways involved in angiogenesis (Figure 6C–G). In addition, all treatments significantly upregulated pathways involved in leukocyte migration and cell chemotaxis. These results, when compared to those presented by naïve treatment, suggested that TME remodeling was highly related to downregulation of angiogenesis and differential regulation of leukocyte migration in this animal model. Furthermore, except for the anti-CTLA-4 treatment, all second-line treatment regimens effectively increased levels of the IFN pathway signature (Appendix AB–F). Therefore, GSEA was performed using the published gene set for the TIL signature. In anti-CTLA-4 antibody-treated tumors, the neutrophil gene signature was upregulated, whereas the IFN pathway and NK cell gene signatures were downregulated (Appendix AB; *p* < 0.05). In CC-02 treated tumors, we observed upregulated IFN pathway, macrophage, NK, and T-cell gene signatures (Appendix AC; *p* < 0.05). In anti-CTLA-4 + CC-02 treated tumors IFN pathway, macrophage, NK, neutrophil gene and T-cell gene signatures were upregulated (Appendix AD; *p* < 0.05). In anti-CTLA-4 + cabozantinib + CD-k30 (chidamide-k30) treated tumors the IFN pathway, macrophage, neutrophil, NK, monocyte, and T-cell gene signatures were upregulated (Appendix AE; *p* < 0.05). Anti-CTLA-4 + regorafenib + CD-k30 treated tumors showed downregulation of macrophage, monocyte, and T-cell gene signatures and upregulation of IFN pathway and neutrophils (Appendix AF; *p* < 0.05). These findings demonstrated that the intratumor heterogeneity of tumor responses to double and triple combinations could limit the predictive outcomes of these signatures. Although anti-CTLA-4 antibody can activate neutrophil gene expression, it significantly suppresses the IFN pathway and NK cell gene expression, which might exert synergistic antitumor gene expression when in combination with other drug(s). CC-02 significantly activated the IFN pathway, and macrophage, NK and T-cell gene expressions, and had no effect on neutrophil gene expression. In addition, both CC-02 and anti-CTLA-4 antibody + CC-02 regimens significantly activated the IFN pathway, and macrophage, NK and T-cell gene expressions, while also reflecting a similar antitumor activity as shown in Figure 5G. In addition, treatment with anti-CTLA-4 antibody + cabozantinib + chidamide-k30 significantly activated NK and T-cells, and monocytes, as well as macrophage and neutrophil gene expressions, while treatment with anti-CTLA4 antibody + regorafenib + chidamide-k30 activated neutrophil gene expression and significantly suppressed macrophage, monocyte, and T-cell gene expressions, while both regimens reflected a similar antitumor activity, as shown in Figure 5G. In the TME, the antitumor mechanisms underlying these two triple combination regimens may be different and further elucidation in future study is necessary.

Previously [28], we proposed that anti-PD-1 resistance (non-responsive to first-line anti-PD-1 Ab treatment as shown in Figure 5B) is due to loss of antigen presentation, as well as downregulation of cell chemotaxis, the IFN pathway signature, and T-cell-mediated cytotoxicity (data also shown in Appendix AA; *p* < 0.05). As shown in Appendix AB, our data revealed that CC-02, anti-CTLA-4 antibody + CC-02, and anti-CTLA-4 antibody + cabozantinib + chidamide-k30 treatment could upregulate transcripts associated with antigen processing and presentation of peptide antigens, and activate T-cell-mediated cytotoxicity in PD-1-resistant tumors. These results suggested that overcoming anti-PD-1 resistance may be achieved via enhanced T-cell activation through the recovery of the antigen presentation pathway and TME modulation. Collectively, the regulation of gene expression described above implied that for effectively overcoming first-line anti-PD-1 antibody-induced resistance, TME regulation should involve gene expression in immune cells, in CT26 tumors. Overcoming resistance to anti-PD-1 antibody therapy as first-line therapy is an urgent clinical issue. Our study suggested that chidamide combined with regorafenib/cabozantinib plus anti-CTLA-4 antibody is an important combination therapy to address resistance induced by first-line anti-PD-1 antibody therapy.

## 3. Discussions

In the present study, we used a novel treatment strategy that combined chidamide (epigenetic immunomodulator) with TKIs and ICI antibody for TME remodeling, resulting in enhanced antitumor responses in CT26-bearing mice. The triple combination regimen chidamide + VEGFR-TKI (cabozantinib or regorafenib) + anti-PD-1 antibody allowed potent eradication of primary tumors by enhancing CD8+ T-cell infiltration and reducing PMN-MDSCs and TAMs (Figure 2). Additionally, we demonstrated that the triple combination is more effective when comprising ICI anti-PD-L1 or anti-CTLA-4 antibody other than anti-PD-1 antibody, resulting in improvement of antitumorigenic response, immune memory, and OS (Figure 4). Furthermore, regarding drug resistance caused by first-line treatment with anti-PD-1 antibody, the second-line treatment with anti-CTLA-4 + cabozantinib/regorafenib + chidamide exerted superior antitumor responses to overcome drug resistance (Figure 5). Finally, the induction of immune activation, as determined by the RNA-seq assay, was identified following second-line treatment with the triple combination regimen, indicated by activation of IFN-γ response, cell chemotaxis, and suppression of the angiogenesis gene signature (Figure 3 and Figure 6). In addition, we identified that the triple regimen of anti-CTLA-4 antibody + regorafenib + chidamide induced the suppression of macrophage and monocyte gene signatures to modulate TME (Appendix A) and ameliorated anti-PD-1 resistance by activating the T-cell-mediated cytotoxicity gene signature (Appendix A).

Regorafenib was shown to increase OS when used as a single-agent therapy in patients with advanced colorectal cancer who had previously failed to respond to chemotherapy regimens, and was approved by the FDA in 2012 [31]. Regorafenib targets multiple receptor tyrosine kinases, including those involved in tumor angiogenesis (VEGFR-1, -2, -3, endothelial-specific receptor tyrosine kinase 2 (TIE2)) and tumor immunity (Colony stimulating factor 1 receptor (CSF1R)) [32]. Regorafenib is a potent, orally-administered, multiple TKI, approved by the US FDA for several indications, including mCC, gastrointestinal stromal tumors, and HCC, and possesses potent immune regulatory properties. Cabozantinib inhibits tumor angiogenesis while targeting the hepatocyte growth factor receptor protein (MET), which modulates tumor immunity [33]. Based on evidence from the pivotal RESORCE (NCT01774344) and CELESTIAL (NCT01908426) phase 3 trials, current HCC management guidelines recommend either regorafenib or cabozantinib for advanced disease after progression, following sorafenib therapy [34,35]. Cabozantinib is a potent multiple TKI approved by the US FDA for several indications, including medullary thyroid cancer, advanced renal cell carcinoma (RCC), and hepatocellular carcinoma (HCC).

Despite employing regorafenib or anti-PD-1 antibody as third-line treatment in advanced colon cancer therapy, no robust and durable clinical responses were documented when used as single agents in proficient mismatch repair (pMMR)/microsatellite stable (MSS)-colorectal cancer. As previously reported, CT26 tumors have a lower mutational load and exhibit characteristics approximating those of MSS/pMMR colorectal cancer, showing poor response to anti-PD1 antibody monotherapy [36,37,38]. Combinations of both anti-angiogenesis drugs and immunotherapies are currently being explored to improve therapeutic outcomes in patients with advanced colorectal cancer. The efficacy could be further improved; however, combination therapy with regorafenib and anti-PD-1 antibody has been unsuccessful in clinical trials. This finding indicates that such a combination is inadequate and cannot adequately address the therapeutic needs of MSS-type colorectal cancer. In our study, data demonstrated that regorafenib combined with an anti-PD-1 antibody regimen was insufficient to boost the tumor response rate; the addition of chidamide (a potent epigenetic immunomodulator) was needed for TME remodeling and marked improvement in the response rate, as shown in Figure 1 and Figure 2. A previous preclinical study highlighted the synergistic immunomodulatory effects of regorafenib and anti-PD1 antibody combination therapy for inducing sustained M1 polarization, as well as sustained reduction of Tregs, which can explain the prolonged CT26 colon tumor suppression [39]. Although tumor growth was inhibited in each mouse, ORR data were not analyzed in this study, which may be attributed to the lack of CR or PR, as shown in Figure 1E. Herein, regorafenib + anti-PD1 antibody combination regimen failed to significantly prolong survival time in CT26-bearing mice, as shown in Figure 1F. However, the regorafenib + anti-PD1 antibody + chidamide-k30 regimen achieved 30–43% ORR and markedly prolonged survival time, suggesting that chidamide is a critical component for improving the anti-PD-1 antibody + regorafenib regimen to achieve better efficacy in terms of ORR and survival rate, as shown in Figure 1E,F. After treatment discontinuation, the antitumorigenic effects of regorafenib + anti-PD-1 antibody therapy were rapidly abrogated; however, the combination regimen of regorafenib +anti-PD-1 antibody + chidamide not only significantly prolonged the survival time and led to sustained suppression of relapse, but also resulted in induction of immunity for the tumor growth inhibition of secondary CT26 cell inoculation in a rechallenge test (Table 1 and Table 2). Likewise, treatment with cabozantinib + anti-PD1 antibody, with or without chidamide regimens, could afford similar results in terms of boosting the efficacy of ORR and survival rate. These results suggest, for the first time, that regorafenib/cabozantinib + chidamide + anti-PD1 antibody regimens could exert a durable tumor-specific response, along with markedly increased ORR and survival rate in CT26-bearing mice.

Regorafenib, cabozantinib, and chidamide monotherapy partially inhibited tumor growth during continued treatment in CT26-bearing mice. ICIs (anti-PD-1, anti-PD-L1, and anti-CTLA-4 antibodies) + regorafenib/cabozantinib + chidamide exhibited a synergistic effect, inducing immune memory in the CT26-bearing mouse model and prolonging survival. These results were in line with a previous study, which reported that a combination of trametinib with immunomodulatory agents, targeting PD-1, PD-L1, or CTLA-4, was more efficacious than any single agent in a CT26 model [40]. To investigate the anticancer mechanisms after treatment with regorafenib/cabozantinib + anti-PD-1 antibody + chidamide, we analyzed changes in immune cell composition (flow cytometry) and immune regulatory genes (RNA-seq and GSEA). In our current study, regorafenib or cabozantinib + anti-PD-1 antibody, with or without chidamide treatment, significantly reduced Tregs in tumors, suggesting that the anticancer activity may be attributed to Treg suppression, as shown in Figure 2F. These findings were consistent with a previous study showing that VEGF binds to the VEGF co-receptor neuropilin1 in Tregs, which is critical for tumor homing [41], and cabozantinib decreased Treg-mediated tumor infiltration, which would improve the T-cell immune response [42]. Compared with IgG, the anti-PD-1 antibody + regorafenib + chidamide group showed higher intratumoral levels of CD8+ T-cells and reduced intratumoral levels of CD11b+ cells, PMN-MDSCs, and TAMs (Figure 2E,G,H,J). These results indicated the potential of the chidamide plus regorafenib regimen as a TME regulator, reducing suppressor cells and favoring an environment for CD8+ T-cells to activate immunity, thereby suppressing tumor growth. These results were in line with previous studies that demonstrated that regorafenib/anti-PD-1 antibody combination therapy could inhibit tumor growth and increase survival by normalizing tumor vasculature and enhancing intratumoral CXCR3+ CD8+  T-cell infiltration in HCC cells [43], and regorafenib inhibits the recruitment of TAMs and Treg cells in tumors [44,45,46]. As a similar finding, we observed that the anti-PD-1 antibody + regorafenib + chidamide combination reduced intratumoral levels of PMN-MDSCs and TAMs, as shown in Figure 2H,J. This phenomenon was attributed to the presence of chidamide as shown in our study (regorafenib + anti-PD-1 vs. regorafenib + anti-PD-1 + chidamide), which improved anti-PD-1 antibody + regorafenib-induced tumor suppression.

We found that downregulated gene expression of leukocyte migration, cell chemotaxis, and macrophage gene sets associated with treatment with anti-PD-1 antibody + regorafenib + chidamide regimen, resulted in decrease of myeloid-derived PMN-MDSCs and TAMs in TME, and showed a significant prolonged anticancer activity after last drug administration in CT26-bearing mice, as shown in Figure 1, Figure 2, and Table 1. In addition, the addition of chidamide to the anti-PD-1 antibody + regorafenib combination markedly enhanced upregulation of T-cell-mediated cytotoxicity and interferon pathway signature, as shown in Figure 3 and Appendix A. Furthermore, chidamide enhanced anti-PD-1 antibody + cabozantinib-mediated reduction in Tregs; however, elevating M-MDSCs in the TME, as shown in Figure 2. Tregs and M-MDSCs are immune-suppressive cells. GSEA revealed that addition of chidamide to the anti-PD-1 antibody + cabozantinib combination could enhance upregulation of leukocyte migration and cell chemotaxis, as shown in Figure 3D,E, and upregulation of the macrophage gene set, neutrophil gene set, and T-cell gene set, as shown in Appendix AD,E. Comparing differential gene expression induced by anti-PD-1 antibody + regorafenib + chidamide and anti-PD-1 antibody + cabozantinib + chidamide treatments, the TME modulating effect of regorafenib exhibited by cabozantinib, in terms of leucocyte migration and cell chemotaxis, as shown in Figure 3E,G, and macrophage and T-cell gene sets, as shown in Appendix AE,G, showed upregulation by cabozantinib but downregulation by regorafenib. In addition, gene expression level related to angiogenesis was downregulated following both regorafenib and cabozantinib treatment (Figure 4). The normalization of tumor vessels and amelioration of the hypoxic environment by treatment with regorafenib or cabozantinib are important factors regulating tumor immune activity. Therefore, our data suggested and supported the rationale that chidamide plus anti-PD-1 antibody, combined with an anti-angiogenesis drug regimen, possessed potent TME modulatory activity to boost the tumor response rate of CT26 tumor-bearing mice.

Finally, a previous study reported considerable heterogeneity in response to anti-PD-1 antibody immunotherapy in mice, similar to that observed in patients undergoing clinical therapy [47]. In first-line anti-PD-1 antibody therapy-induced resistance, although the anticancer efficacy ORR of second-line therapy with anti-CTLA-4 antibody did not differ from that with anti-PD-1 antibody, the percentage of PD (progressive disease) was lower following anti-CTLA-4 antibody therapy than that in anti-PD-1 antibody treatment, as shown in Figure 5G. CC-02 or any triple combination shown in Figure 5 significantly enhanced the anticancer efficacy when compared to anti-CTLA-4 antibody alone and prolonged OS. In HPD mice, large tumors treated with the triple regimen comprising anti-CTLA-4 antibody + cabozantinib + chidamide-HCl showed an enhanced tumor response rate and prolonged survival time, as shown in Figure 5G and Figure 6H. We observed that this treatment regimen could prevent relapse and inhibit CT-26 tumor growth in HPD mice in a rechallenge test, as shown in Table 4. These results indicated that second-line treatment with any of the triple combinations mentioned in Figure 5 induced a prolonged tumor-specific response following development of resistance to first-line treatment with anti-PD-1 antibody. We identified a signature consisting of the downregulation of angiogenesis by treatment with CC-02 or triple combinations anti-CTLA-4 + regorafenib/ cabozantinib + chidamide, as shown in Figure 6. Additionally, we identified a signature consisting of an upregulation of both response to INF-γ and cellular response to INF-γ in CC-02 and all the triple combination-treated tumors from mice with primary resistance. The efficacy of CC-02 or CC-02 + anti-CTLA-4 antibody was lower than that of anti-CTLA4 antibody + anti-angiogenesis agents + chidamide in treating first-line anti-PD-1 antibody-induced resistance in mice, implying that anti-CTLA4 antibody + anti-angiogenesis agents + chidamide may be a better choice when facing the drug resistance issue caused by first-line anti-PD-1 antibody therapy.

Resistance to tumor immunotherapy can be attributed to a lack of tumor-associated antigens, resulting in lower T-cell priming, weak tumor infiltration [11], and impaired DC maturation [13,14,48]. Compared with naïve mice, anti-PD-1 resistant mice showed downregulated adaptive immune response, antigen processing and presentation, cell chemotaxis, IFN pathway signature, and T-cell-mediated cytotoxicity, as shown in Appendix AA. CC-02 or most triple combinations could significantly recover this effect, given antigen processing and presentation and T-cell-mediated cytotoxicity, as shown in Appendix AB. Although the triple combination anti-CTLA4 antibody + regorafenib + chidamide did not show significant effect on antigen presentation (Appendix AB), it could enhance the tumor response rate and prolong survival time, which was coincident with the observation of upregulation of T-cell-mediated cytotoxicity. These results suggested that anti-CTLA4 antibody + anti-angiogenesis drugs + chidamide might overcome anti-PD-1 antibody resistance by stimulating cytotoxic T-cell activation. This finding was consistent with a previous study on epigenetic modulation via DNA methyltransferase and HDACis that rationally sensitized tumors to anti-PD1/PD-L1 antibody therapy [49]. Despite the different targets of regorafenib and cabozantinib, the same high response rate of regorafenib/cabozantinib + anti-PD-1 Ab + chidamide was observed in the CT26-bearing mouse model. However, combining regorafenib prolonged survival time in both naïve and anti-PD-1 antibody-resistant mice (as shown in Figure 1, Figure 2 and Figure 6I), accompanying the observation that regorafenib induced substantially more immune activation, possibly by suppressing PMN-MDSCs and TAMs in tumors (Figure 2H,J).

The limitations of the present study need to be addressed. First, it remains unclear what kind of cytokines or chemokines affect cell homing to the tumor by anti-angiogenesis drugs + anti-PD-1 antibody + chidamide treatment, which affects the composition of immune cells in the TME. Second, few reports have focused on the immune-related hematologic adverse drug events (Hem-irAEs) of ICIs in patients with cancer [50], and reported adverse events rarely have highlighted myelosuppression [31,51]; only chidamide has reported significant hematologic events [52]. From our study we did not acquire enough evidence to make any conclusion about the correlation between immune cell migration/abundance and immune cell components, especially regarding immune-suppressive cells in the TME (as shown in Appendix A and Figure 2). Therefore, we postulate that the decrease in myeloid cells in tumors may be due to chidamide-induced suppression of myeloid cell homing and, therefore, to increase in the anticancer activity of the ICI + anti-angiogenesis drug. In addition to regulating the TME, it is crucial to consider the issue of the bone marrow microenvironment, as all immune cell progenitors are derived from the bone marrow. The contribution of chidamide to regulating the bone marrow and TME may underlie the success of this triple-drug combination regimen.

## 4. Material and Methods

### 4.1. Anti-Colorectal Cancer Activity in Animal Models

The animal study was approved and performed under the guidance of the Taipei Medical University Institutional Animal Care and Use Committee (TMU IACUC, NO: LAC-2020-0103, LAC-2019-0644). Six- to eight-week-old male BALB/c mice (National Laboratory Animal Center, Taiwan) were used to perform experiments. The CT26 cell line was purchased from American Type Culture Collection (ATCC, Manassas, VA, USA). CT26 tumor cell lines were grown in RPMI-1640 supplemented with 10% (*v*/*v*) fetal bovine serum at 37 °C and 5% CO_2_. For establishing tumors, a 1 × 10^6^ CT26 cell suspension in 100 µL phosphate-buffered saline (PBS) with 50 µL Matrigel (354248, Corning^®^ BD Biosciences, Corning, NY, USA) was subcutaneously injected into the left flank of the mice, and tumor growth was determined by measuring two perpendicular diameters. Prior to randomization and treatment, tumors were allowed to grow for 8–12 days (tumor size, approximately 110–250 mm^3^). An anti-IgG antibody (Catalog #BE0089, BioXcell, West Lebanon, NH, USA), anti-PD-1 antibody (BE0146, BioXcell, West Lebanon, NH, USA), anti-PD-L1 antibody (BE0101, BioXcell, West Lebanon, NH, USA), and anti-CTLA-4 antibody (BE0164, BioXcell, West Lebanon, NH, USA) were administered intraperitoneally (i.p.) at 2.5 mg/kg, every 3 days for a total of 6 injections in 16 days. All antibodies were diluted to appropriate concentrations in 100 μL sterile PBS (pH 7.4; Invitrogen Life Technologies, Carlsbad, CA, USA). Axitinib (HY-10065, 30 mg/kg, per oral [p.o.] daily; MedChemExpress, Monmouth Junction, NJ, USA), lenvatinib (HY-10981, 10 mg/kg, p.o. daily; MedChemExpress)cabozantinib (HY-13016, 30 mg/kg, p.o. daily; MedChemExpress), regorafenib (HY-1031, 30 mg/kg, p.o. daily; MedChemExpress), chidamide-k30 or chidamide-HCl salt (50 mg/kg, p.o. daily; produced by GNTbm, Taipei, Taiwan), celecoxib (50 mg/kg, p.o. daily; capsule/Celebrex^®^; Pfizer Inc., New York, NY, USA) were administered for 16 days. Axitinib, lenvatinib, cabozantinib, regorafenib, and celecoxib were dissolved in dimethyl sulfoxide and diluted in water before administration. Chidamide-k30 and chidamide-HCl salts were dissolved in water. Animals were euthanized when tumors exceeded 3000 mm^3^ in volume. The anticancer activity was measured from treatment initiation until the tumor volume reached 3000 mm^3^. The tumor volume was calculated as length × width^2^ × 0.5. For assessing treatment efficacy, we defined complete response (CR; <0.5-fold tumor growth in tumor-bearing mice at three days after the end of treatment), partial response (PR; tumor size ≥ 0.5-fold time tumor growth, but <1-fold tumor growth in the tumor-bearing mice 3 days after the end of treatment), stable disease (SD; tumor size ≥ 1-fold tumor growth, but <5-fold tumor growth in the tumor-bearing mice 3 days after the end of treatment), and progressive disease (PD; tumor size ≥ 5-fold tumor growth in tumor-bearing mice 3 days after the end of treatment). ORR was defined as percentage of mice with CR or PR assessed after treatment in a treatment group. Recurrence was defined as tumor growth of at least 5-fold in mice with a CR or PR response after the first tumor assessment.

### 4.2. Tumor Rechallenge in Animal Models

All mice with a PR/CR response after treatment were rechallenged with CT26 cells on the contralateral side. The CT26 rechallenge was performed on day 34 ± 2, i.e., 7 days after the first tumor assessment (day 27 ± 2), by administering an injection of 5 × 10^6^ CT26 cells per mouse. After rechallenge with CT26 cells, tumors were allowed to grow for another 7 days (day 41 ± 2) to determine the tumor baseline as 1-fold. After an additional 10 days (day 51 ± 2), tumor growth was evaluated for rechallenge. If two of the following criteria were met, the response was considered rechallenge-induced recurrence/relapse: (1) the tumor size was >2-fold when compared to that at baseline on day 41 ± 2; (2) the tumor volume on day 51 ± 2 was >300 mm^3^. A relapse occurs when immunity is insufficiently activated. If the growth of tumor from the second time cancer cell inoculation (the rechallenge) was inhibited, the immune system was activated by the treatment.

### 4.3. Survival Rate in Animal Models

After tumor assessment, the tumor volume of mice was measured once every three or four days (twice weekly). Tumor-bearing mice were considered dead when the tumor volume reached 3000 mm^3^. All treatment groups were recorded and analyzed.

### 4.4. Flow Cytometry

The following antibodies and reagents were used for flow cytometry: CD8a PerCP-Cy5.5 (53–6.7; BioLegend, San Diego, CA, USA), CD4 PE (GK 1.5; BioLegend, San Diego, CA, USA), CD25 PerCP-Cy5.5 (PC61; BioLegend, San Diego, CA, USA), Foxp3 PE (MF14; BioLegend, San Diego, CA, USA), CD3 APC (17A2; BioLegend, San Diego, CA, USA), CD11b APC (M1/70; BioLegend, San Diego, CA, USA), Ly-6C PerCP-Cy5.5 (HK 1.4; BioLegend, San Diego, CA, USA), Ly-6G PE (1A8; BioLegend, San Diego, CA, USA), MHC-II-PE (BM8; BioLegend, San Diego, CA, USA), and CD45 FITC (30-F11; BioLegend, San Diego, CA, USA). Flow cytometry was performed using FACSCaliber (BD Biosciences, San Jose, CA, USA), and data were analyzed using the FACS Diva software (BD Biosciences, San Jose, CA, USA). Subsequently, cells were fixed, permeabilized with BD Cytofix/Cytoperm (BD Biosciences, San Jose, CA, USA), and stained with an antibody against FOXP antibody.

To assess various tumor-infiltrating immune cells in tumors, further assays were performed to analyze intratumoral CD3+, CD8+, CD4+ T-cells, Tregs, polymorphonuclear (PMN)-MDSCs, monocytic-MDSCs (M-MDSCs), and tumor-associated macrophages (TAMs). Cells were first purified from tumor samples excised from mice on day 20 after 9-day treatment of cabozantinib or regorafenib, with or without chidamide-k30 plus anti-PD-1 antibody. Briefly, primary tumor tissues were harvested, weighed, and minced into fine fragments. Collagenase IV (Sigma-Aldrich, St Louis, MO, USA)) at 1 mg/mL in Hanks’ Balanced Salt Solution (HBSS; Invitrogen Life Technologies, Grand Island, NY, USA) was added to each sample at a ratio of 1 mL per 200 mg of tumor tissue. Samples were incubated on an end-over-end shaker for 150 min at 37 °C. The resulting tissue homogenates were filtered through a 0.4-μm filter and washed three times in PBS (BD Biosciences, San Jose, CA, USA), followed by separation using a Percoll gradient to isolate mononuclear cells, and 1 × 10^6^ cells per sample were used for antibody labeling. CD8+ T-cell levels were assessed using previously established phenotypic criteria for CD45+ CD3+ CD8+. Treg levels were assessed using previously established phenotypic criteria for CD45+ CD4+ CD25+ FoxP3+; PMN-MDSC and M-MDSC levels were assessed using previously established phenotypic criteria for CD45+/CD11b+/Ly6G+/Ly6C− and CD45+/CD11b+/Ly6G−/Ly6C+, respectively. Finally, TAM cell levels were assessed using previously established phenotypic criteria for CD45+ CD11b+ MHC-II+ Ly6C+, and total mononuclear cells were used as a common denominator.

### 4.5. Overcoming Primary Resistance and HPD Induced by First-Line PD-1 Checkpoint Blockade Therapy

Male BALB/c mice bearing subcutaneous CT26 tumors (1 × 10^6^ cells/mouse) were treated with anti-PD-1 antibody as first-line therapy (mean tumor volume: 113 mm^3^ at treatment initiation), administered i.p. at 2.5 mg/kg, once every 3 days for 3 doses. Treatment was continued for an additional 3 doses (i.e., a total of 6 doses), if tumors responded to anti-PD-1 antibody treatment (tumors shrunk or with a growth of less than 2.5-fold; please also refer to the definition described below). Acquired resistance was defined as tumors that shrunk after the first 3 doses of anti-PD-1 antibody therapy, then grew gradually following continuous anti-PD-1 antibody treatment (i.e., a total of six doses), presenting only partially inhibited tumor growth and subsequently exhibiting further growth. Primary resistance was defined as tumor failure to respond to anti-PD-1 antibody therapy (after three doses), with a 2.5-fold increase in tumor volume but <600 mm^3^. HPD was defined as tumors that grew >600 mm^3^ after three doses of first-line anti-PD-1 antibody treatment. For the efficacy study, mice with primary resistance, acquired resistance, and HPD were subsequently re-enrolled for second-line therapy, as shown in Figure 5A. Second-line therapies used were as follows: anti-IgG antibody as a control; anti-PD-1 antibody, and anti-CTLA-4 antibody administered i.p. at 2.5 mg/kg, once every 3 days for 6 doses; chidamide-HCl (50 mg/kg) + celecoxib (50 mg/kg) (CC-02), regorafenib (30 mg/kg) + chidamide-k30 (50 mg/kg), and cabozantinib (30 mg/kg) + chidamide-k30 (50 mg/kg), orally administered once daily for 16 days. Tumor length and width was measured once every three or four days (twice weekly) by using a caliper. The anticancer activity was measured from treatment initiation until the tumor volume reached 3000 mm^3^. Tumor volume was measured as described above.

### 4.6. RNA Quantification and Qualification

Naïve CT26 tumor-bearing mice were randomized and treated with different regimens, and tumors were excised and collected on day 20 after 9-day therapy. In contrast, following first-line anti-PD-1 antibody therapy, drug-resistant mice were randomized and treated with different regimens, and the tumors were excised and harvested on day 28 after initiating second-line therapy. All tumor samples were snap-frozen in liquid nitrogen and homogenized in Trizol (Life Technologies, Carlsbad, CA, USA). RNA purity and quantification were assessed using the SimpliNano™-Biochrom spectrophotometer (Biochrom, Holliston, MA, USA). RNA degradation and integrity were examined using a Qsep 100 DNA/RNA analyzer (BiOptic Inc., new Taipei City, Taiwan).

### 4.7. Library Preparation for Transcriptome Sequencing

Total RNA (1 μg) per sample was used as input material for RNA sample preparation. Sequencing libraries were generated using the KAPA mRNA HyperPrep Kit (KAPA Biosystems, Roche, Basel, Switzerland), following the manufacturer’s recommendations, and index codes were added to attribute sequences to each sample. PCR products were purified using the KAPA Pure Beads system, and library quality was assessed using a Qsep 100 DNA/RNA Analyzer (BiOptic Inc., new Taipei City, Taiwan).

### 4.8. Bioinformatics

The original data obtained by high-throughput sequencing (Illumina NovaSeq 6000 platform) were transformed into raw sequenced reads by CASAVA base calling and stored in the FASTQ format. FastQC and MultiQC were used to examine the quality of the FASTQ files. The obtained raw paired-end reads were filtered using Trimmomatic (v0.38) to discard low-quality reads, trim adaptor sequences, and eliminate poor-quality bases using the following parameters: LEADING:3 TRAILING:3 SLIDINGWINDOW:4:15 MINLEN:30. The obtained high-quality data (clean reads) were used for subsequent analysis. Read pairs from each sample were aligned to the reference genome using HISAT2 software (v2.1.0). FeatureCounts (v1.6.0) were used to count read numbers mapped to individual genes. For gene expression, normalization of “Trimmed Mean of M-values” (TMM) was performed using DEGseq (v1.36.1) without biological duplicates, while normalization of “Relative Log Expression” (RLE) was performed using DESeq2 (v1.22.1) with biological duplicates. Differentially expressed gene (DEG) analysis for the two conditions was performed in R using DEGseq (without biological replicates) and DESeq2 (with biological replicates), which were based on negative binomial distribution and Poisson distribution models, respectively. The resulting *p*-values were adjusted using Benjamini and Hochberg’s approach for controlling the false discovery rate. Gene Ontology (GO) and Kyoto Encyclopedia of Genes and Genomes (KEGG) pathway enrichment analysis of DEGs were conducted using clusterProfiler (v3.10.1). Gene set enrichment analysis (GSEA) was performed with 1000 permutations to identify enriched biological functions and activated pathways from the molecular signature database (MSigDB). MSigDB is a collection of annotated gene sets for use with GSEA software, including hallmark, positional, curated, motif, computational, GO, oncogenic, and immunologic gene sets. In addition, weighted gene co-expression network analysis (WGCNA) was performed using the co-expression network based on the correlation coefficient of expression patterns using the WGCNA (v1.64) package in R.

### 4.9. Statistics

All statistical analyses were performed using GraphPad Prism 5.0 (GraphPad Software, Inc., La Jolla, CA, USA). Primary tumor growth curves, flow cytometric analyses, and anti-PD-1-treated resistance tumor growth curves were first analyzed with one-way ANOVA, and individual groups were compared using Tukey’s Multiple Comparison Test. Kaplan–Meier survival curves were analyzed with a log-rank test.

## 5. Conclusions

The present study showed that in naïve or first-line anti-PD-1 antibody resistant CT26 tumor-bearing mice, an epigenetic immunomodulator drug, i.e., chidamide, combined with ICIs + anti-angiogenesis drug, markedly increased the tumor response rate. In addition, we demonstrated that triple combination therapy induced a sustained anticancer response, prevented relapse, and boosted immunity after treatment cessation. Furthermore, chidamide + anti-angiogenesis drug combined with other ICIs could improve the response rate, while also suggesting solution for overcoming anti-PD-1 antibody resistance. This finding provides rationale for clinical trials to enhance the therapeutic benefits of cancer immunotherapy and effectively overcome the resistance arising from the use of first-line ICIs.

## Figures and Tables

**Figure 1 ijms-23-10677-f001:**
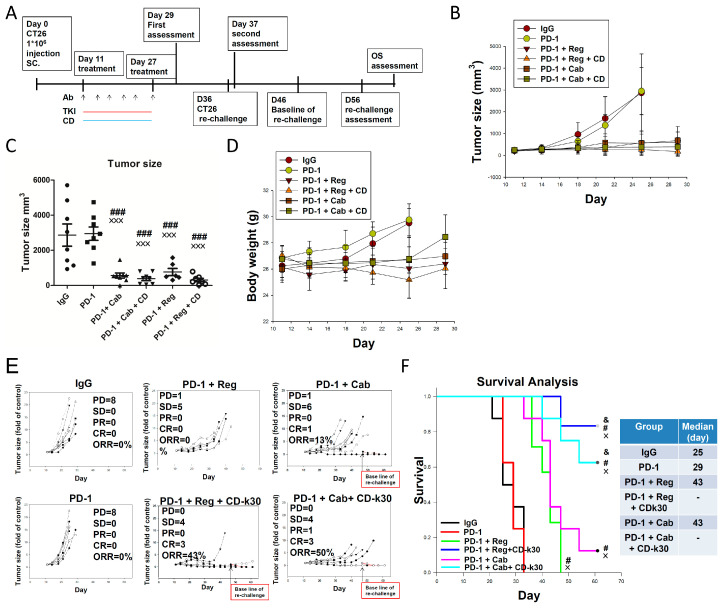
(**A**–**F**) shows the results of therapeutic responses to Cabozantinib or Regorafenib combined with Chidamide-k30 plus anti-PD-1 antibody in CT26 tumor-bearing mice. Balb/c mice bearing a CT26 tumor were treated with various therapeutic modalities as indicated. The combination with Cabozantinib/ Regorafenib is shown in (**A**) consecutive treatment schedule (**B**) total tumor volumes (**C**) Endpoint evaluated tumor volumes at D25 (**D**) Mice body weights (**E**) Individual tumor volumes (**F**) animal survival rates. CT26 tumor-bearing mice were treated as indicated and euthanized at a tumor volume of 3000 mm^3^ after tumor implantation. After 16-day treatment followed by tumor assessment, the mice with CR and PR response were re-inoculated with CT-26 cancer cells into the opposite flank in a rechallenge experiment. Red line squares indicates re-inoculated CT26 tumor growth in rechallenge experiment. Data are given as mean ± SD; × *p* < 0.05, ××× *p* < 0.001, ### *p* < 0.001, one-way ANOVA with Tukey’s test. ×, compared to IgG; #, compared to PD-1, & compared to PD-1 + Reg. (*n* = 6–8).

**Figure 2 ijms-23-10677-f002:**
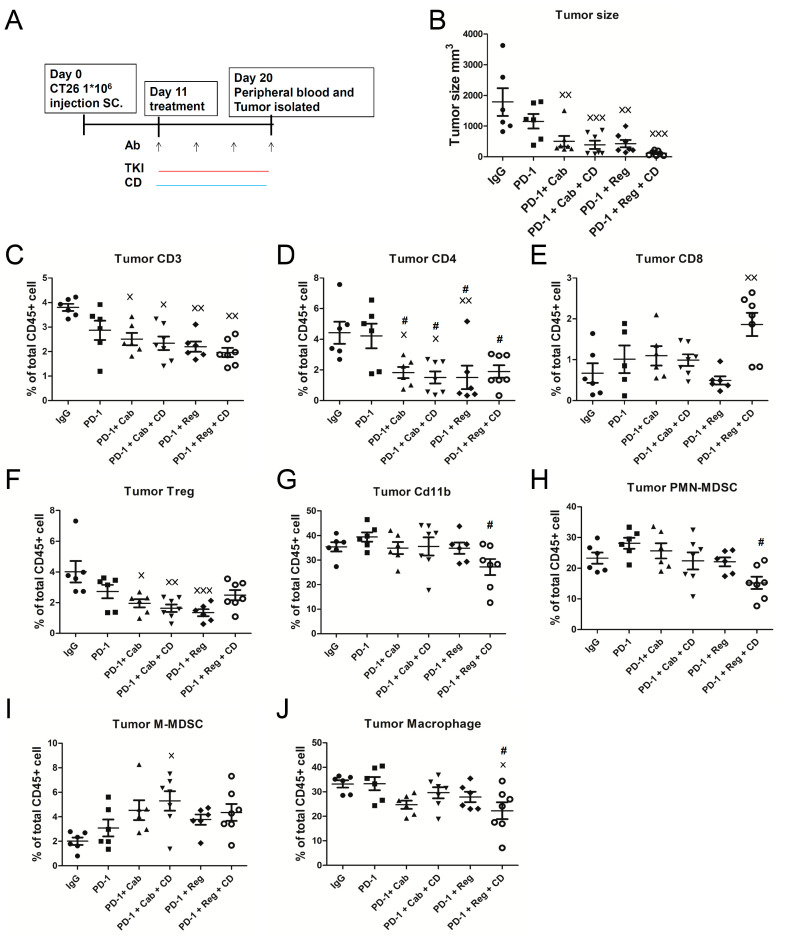
Immunosuppressive cells in microenvironment were attenuated by anti-PD-1 +TKI +CD-k30 treatment. (**C**–**J**) show the results of immune cell population analysis of lymphocytes and myeloid-derived MDSCs in the CT26-bearing mice tumors. The CT26 tumor-bearing mice were treated with various therapeutic modalities as indicated. Tumor samples were isolated on day 20 after 9-day treatment for analyzing immune cell population in tumors. (**A**) Consecutive treatment schedule and (**B**) Tumor sizes of each treatment group. (**C**–**F**) show the results of flow cytometry of CD3, CD4, CD8, and Treg cell population in tumors. (**G**–**J**) show the results of flow cytometry of myeloid-derived CD11b, PMN-MDSC, M-MDSC, and tumor association macrophage (TAM) cell populations in tumors. Results are shown as mean ± SD; × *p* < 0.05, ×× *p* < 0.01, ××× *p* < 0.001, one-way ANOVA with Tukey’s test. ×, compared to IgG; #, compared to PD-1 (*n* = 6–7).

**Figure 3 ijms-23-10677-f003:**
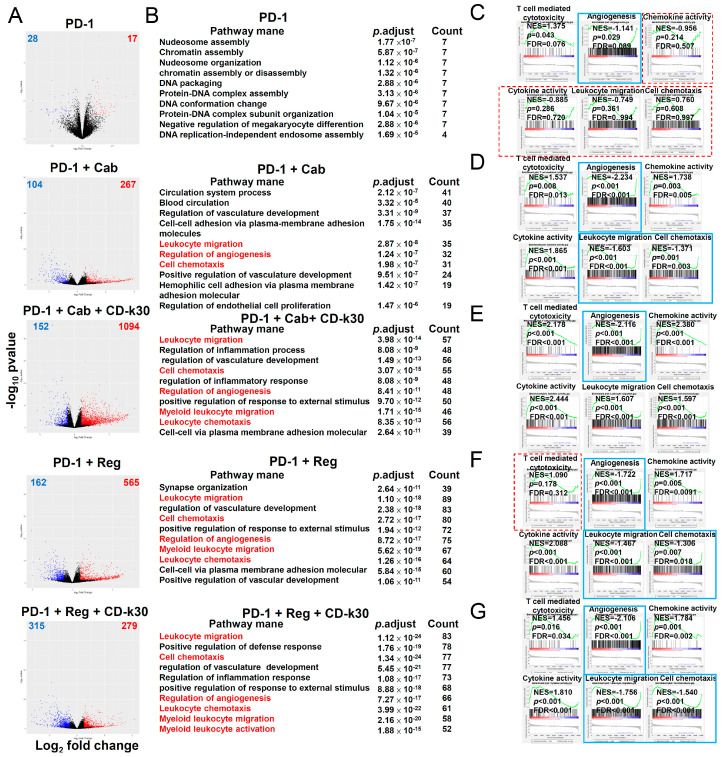
Identification of target genes of anti-PD-1 Ab combined with Regorafenib/Cabozantinib plus Chidamide-k30 treatment, which significantly regulates gene expression in TME of CT26 tumors-bearing mice. Tumors were analyzed on day 20 after 9-day treatment for gene expression by RNA-seq. (**A**) Volcano plot of differentially expressed genes obtained by RNA-seq analysis in anti-PD-1 Ab combined with Regorafenib/Cabozantinib plus Chidamide-k30 treatment CT26 tumors compared to IgG control tumors. Significantly upregulated or downregulated genes are plotted in red and blue points, respectively. (**B**) Meta-enrichment analysis summary for significantly upregulated and downregulated genes was indicated by display of categories of related pathways and the number of affected genes of the corresponding pathway. The pathways highlighted with red color were related to the gene expression signatures in (**C**–**G**). (**C**–**G**) show that gene expression related to chemokine activity, cytokine activity, leukocyte migration, cell chemotaxis, T-cell-mediated cytotoxicity and angiogenesis were analyzed in tumors. (**C**) GSEA enrichment analysis of tumors treated with anti-PD-1. (**D**) GSEA enrichment analysis of tumors treated with PD-1 + cab. **(****E**) GSEA enrichment analysis of tumors treated with anti-PD-1 + Cab + CD-k30. (F) GSEA enrichment analysis of tumors treated with anti-PD-1 + Reg. (**G**) GSEA enrichment analysis of tumors treated with anti-PD-1 + Reg + CD-k30. NES: normalized enrichment score; FDR: false discovery rates. Signature scores were calculated by mean log2 (TPM) of their respective member genes; *p*-values: Mann-Whitney test, two-tailed. When *p* ≧ 0.05, the GSEA analysis panel(s) is outlined with a red dotted line. When gene expression was downregulated, the GSEA analysis panel(s) is outlined with a blue solid line.

**Figure 4 ijms-23-10677-f004:**
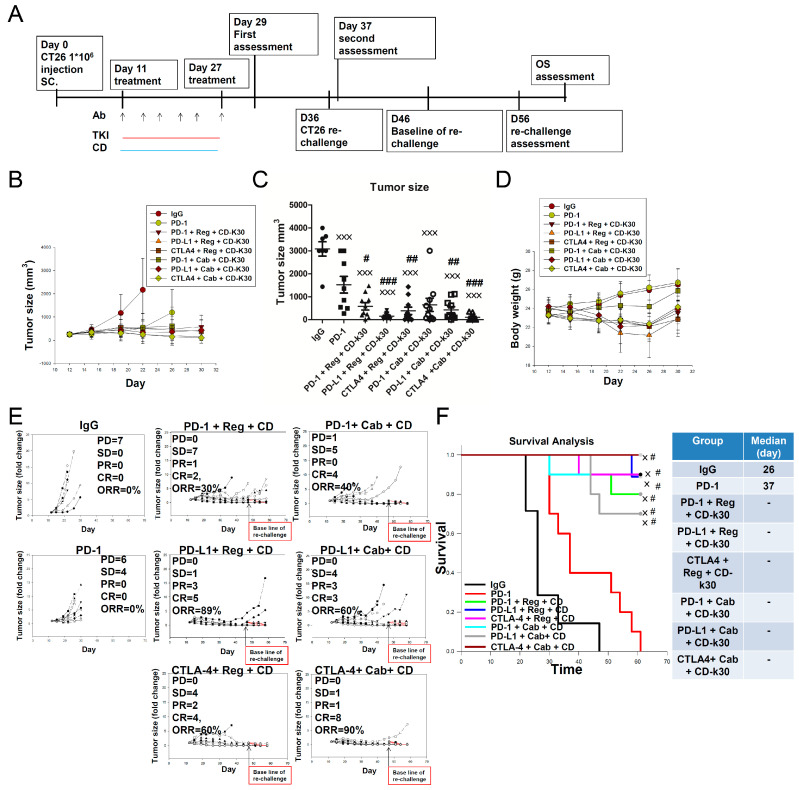
(**A**–**F**) shows the results of therapeutic responses and immunity evaluations of different ICIs combined with Cabozantinib or Regorafenib plus Chidamide-k30 in CT26 tumor-bearing mice. Balb/c mice bearing a CT26 tumor were treated with various therapeutic modalities as indicated. The combination therapy is shown in (**A**) consecutive treatment schedule and (**B**) total tumor volumes. (**C**) Endpoint evaluated tumor volumes at D26. (**D**) Mice body weights. (**E**) Individual tumor volumes. (**F**) Animal survival rates. CT26 tumor-bearing mice were treated as indicated and euthanized at a tumor volume of 3000 mm^3^ after tumor implantation. After 16-day treatment followed by tumor assessment, the mice with CR and PR response were re-inoculated with CT-26 cancer cells into the opposite flank in a rechallenge experiment. Red line squares indicates re-inoculated CT26 tumor growth in rechallenge experiment. Data are given as mean ± SD; × *p* < 0.05, ××× *p* < 0.001 compared to IgG; #, compared to PD-1; one-way ANOVA with Tukey’s test. (*n* = 7–10).

**Figure 5 ijms-23-10677-f005:**
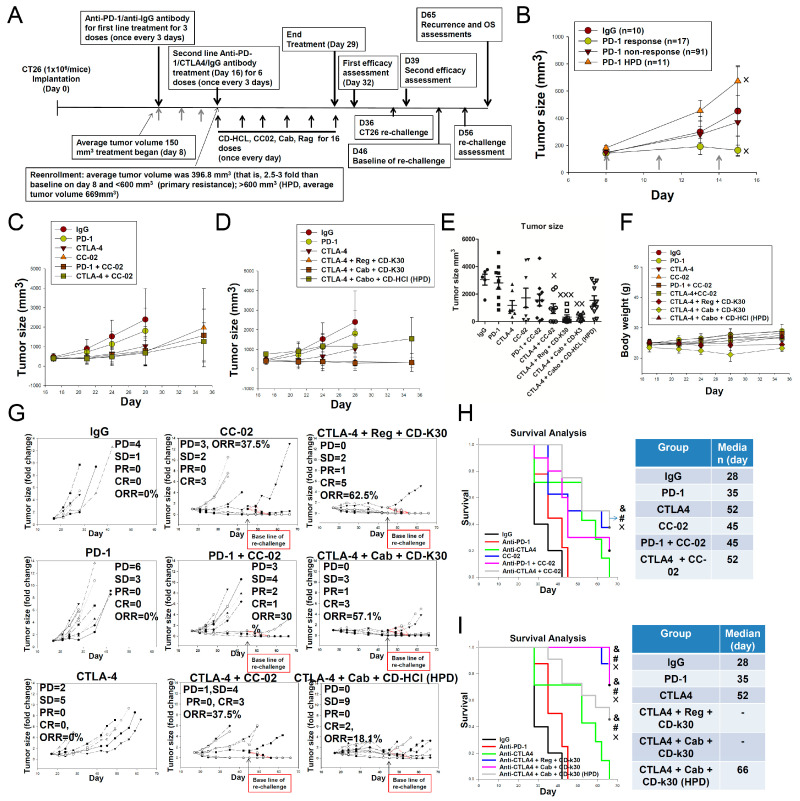
(**A**) shows the consecutive treatment schedule and responsive results of the first-line anti-PD-1 Ab treatment. Male Balb/c mice bearing subcutaneous CT26 tumors (1 × 10^6^ cell/mice) were treated with a first-line therapy of anti-PD-1 Ab (mean tumor volume: 150 mm^3^ when treatment began). The mice were intraperitoneally (i.p.) administered anti-PD-1 Ab or IgG at 2.5 mg/kg, once every 3 days for 3 doses. When the mice responded to anti-PD-1 Ab with tumor shrinking, they were given three more doses. (**B**) Tumor size (mm^3^) from mice responsive to first-line anti-PD-1 Ab treatment, in comparison with control group treated with anti-IgG antibody 3 times. *, *p* < 0.05. Figure 5C–E show the results of a second-line treatment in the mice having anti-PD-1 antibody primary resistance. In the first-line anti-PD-1 antibody therapy, if the tumor showed 2.5- to 3-times consecutive increases in tumor volume and with volumes of <600 mm^3^, the mice were defined as having primary resistance. These mice were subsequently reenrolled and divided into nine groups in a second-line treatment for efficacy study. In the second-line treatment, anti-IgG antibody was as a Ab control, anti-PD-1, and anti-CTLA-4 antibodies were administered intraperitoneally (i.p.) at 2.5 mg/kg, once every 3 days for 6 doses. The combinations in the second-line treatment were: Chidamide-HCL salt (50 mg/kg) + Celecoxib (50 mg/kg) (CC-02), anti-PD-1(2.5 mg/kg) + CC-02, anti-CTLA-4 antibody (2.5 mg/kg) + CC-02 as shown in (**C**) Tumor size (mm^3^) Anti--CTLA-4 antibody (2.5 mg/kg) combined with Regorafenib (30 mg/kg) (reg) plus Chidamide-k30 (50 mg/kg) (CD-k30); and anti-CTLA-4 antibody (2.5 mg/kg) combined with Cabozantinib (30 mg/kg) (cab) plus Chidamide-k30 (50 mg/kg) (CD-k30) as shown in (**D**) tumor size (mm^3^). Figure 1D also shows the results of second-line treatment in mice with hyperprogressive disease (HPD) tumor during anti-PD-1 antibody therapy. After three times of administration of first-line Anti-PD-1 antibody, if the tumor volumes were >600 mm^3^, the mice were defined as having hyperprogressive disease (HPD). These mice were subsequently reenrolled in a second-line treatment for efficacy study. The combination in the second-line treatment is Anti-CTLA-4 antibody (2.5 mg/kg) combined with Cabozantinib (30 mg/kg) plus Chidamide-HCl salt (50 mg/kg). (**E**) Efficacy evaluated endpoint tumor size (mm^3^) at D28, (**F**) Mice body weights. (**G**) show the results of individual tumor volume in the second-line treatment of the anti-PD-1 Ab primary resistance mice. (**H**,**I**) Overall survival rates after second-line treatment for mice with primary resistance to anti-PD-1 antibody. After 16-day treatment followed by tumor assessment, the mice with CR and PR response were re-inoculated with CT-26 cancer cells into the opposite flank in a rechallenge experiment. Red line squares indicates re-inoculated CT26 tumor growth in a rechallenge experiment. Data are given as mean ± SD; × *p* < 0.05, ×× *p* < 0.01, ××× *p* < 0.001 compared to IgG; #, compared to PD-1; &, compared to CTLA-4, one-way ANOVA with Tukey’s test. (*n* = 5–11).

**Figure 6 ijms-23-10677-f006:**
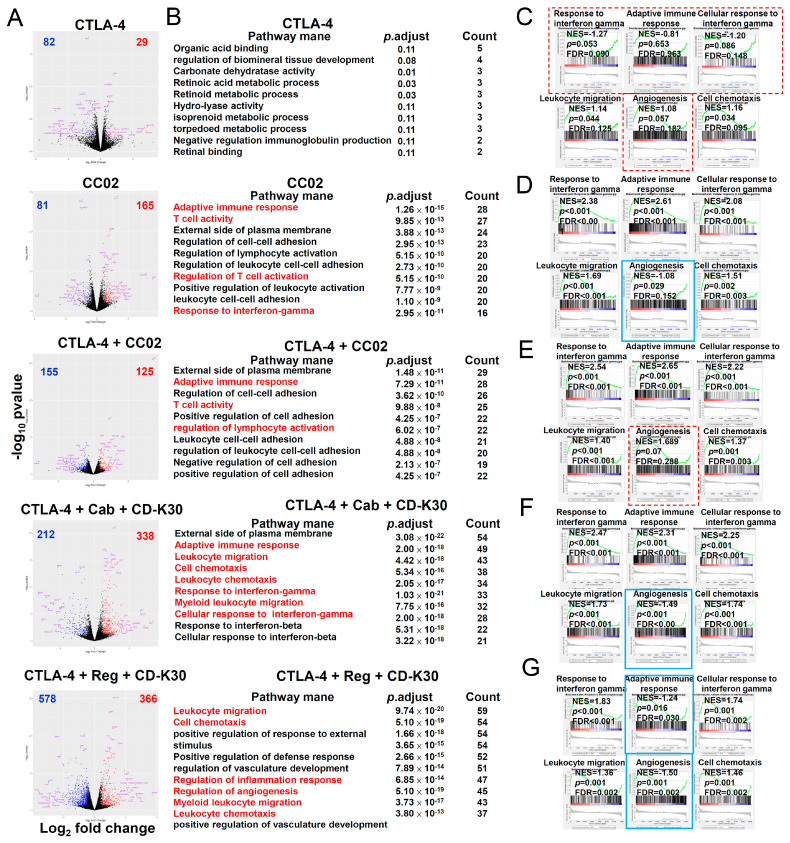
Identification of target genes of second-line treatment with anti-CTLA-4 Ab combined with CC-02 or Regorafenib/Cabozantinib plus Chidamide-k30 that significantly regulates gene expression in TME of CT26 tumor-bearing mice. Tumors were analyzed on day 12 after starting second-line treatment for gene expression by RNA-seq. (**A**) Volcano plot of differentially expressed genes obtained by RNA-seq analysis in treated CT26 tumors compared to IgG control tumors. Significantly upregulated or downregulated genes are plotted in red and blue points, respectively. (**B**) Meta-enrichment analysis summary for significantly upregulated and downregulated genes was indicated by display of categories of related pathways and the number of affected genes of the corresponding pathway. The pathways highlighted with red color were related to the gene expression signatures in (**C**–**G**). (**C**–**G**) show results of gene expression related to response to INF-γ, cellular response to INF-γ, leukocyte migration, cell chemotaxis, adaptive immune response, and angiogenesis being analyzed. (**C**) GSEA enrichment analysis of tumors treated with anti-CTLA-4. (D) GSEA enrichment analysis of tumors treated with CC-02. (**E**) GSEA enrichment analysis of tumors treated with anti-CTLA-4 + CC-02. (**F**) GSEA enrichment analysis of tumors treated with anti-CTLA-4 + Cab + CD-k30. (**G**) GSEA enrichment analysis of tumors treated with anti-CTLA-4 + Reg + CD-k30. NES: normalized enrichment score; FDR: false discovery rates. Signature scores were calculated by mean log2 (TPM) of their respective member genes; *p*-values: Mann-Whitney test, two-tailed. When *p* ≧ 0.05, the GSEA analysis panel(s) is outlined with a red dotted line. When gene expression was downregulated, the GSEA analysis panel(s) is outlined with a blue solid line.

**Table 1 ijms-23-10677-t001:** The efficacy of HDAC inhibitor plus tyrosine kinase inhibitor with ICI in CT26 tumor-bearing mice model.

Regimens	Initial Tumor Volume (mm^3^)	ORR (%)	PD	SD	PR	CR	Survival Rate (%)	Relapse * (Recurrence) (%)	Immunity # (Rechallenge) (%)
anti-PD-1 Ab	227	0	8	0	0	0	0	-	-
anti-PD-1 Ab + reg		13	1	6	0	1	13	0	100
anti-PD-1 Ab + cab		0	1	5	0	0	0	-	-
anti-PD-1 Ab + cab+ CD-k30		50	0	4	1	3	50	50	100
anti-PD-1 Ab + reg + CD-k30		43	0	4	0	3	86	0	100

Abbreviations: Ab, antibody; HDACi, histone deacetylase inhibitor; ICI, immune checkpoint inhibitor; ORR, objective response rate; PD, progressive disease; SD, stable disease; PR, partial response; CR, complete response. * Relapse/recurrence was defined as tumor growth of at least 5-fold in mice with CR or PR after the first tumor assessment. #: Mice resistant to CT26 rechallenge. Response evaluation criteria: fold-change of tumor size compared to baseline. PD: x ≥ 5, SD: 1 ≤ x < 5, PR: 0.5 ≤ x < 1, CR: x < 0.5.

**Table 2 ijms-23-10677-t002:** Anticancer activities of different ICIs combined with TKIs plus HDAC inhibitor in CT26 tumor-bearing mice model.

Regimens	Initial Tumor Volume (mm^3^)	ORR (%)	PD	SD	PR	CR	ORR (%) ^&^	PD ^&^	SD ^&^	PR ^&^	CR ^&^	Survival Rate (%)	Relapse * (Recurrence) (%)	Immunity # (Rechallenge) (%)
anti-PD-1 Ab	243	0	6	4	0	0	0	8	2	0	0	0	-	-
anti-PD-1 Ab + reg + CD-k30		30	0	7	1	2	60	1	3	2	4	80	0	100
anti-PD-L1 Ab + reg + CD-k30		89	0	1	3	5	89	0	1	0	8	89	13	100
anti-CTLA-4 Ab + reg + CD-k30		60	0	4	2	4	80	1	1	1	7	90	0	100
anti-PD-1 Ab + cab+ CD-k30		40	1	5	0	4	60	3	1	1	5	70	0	100
Anti-PD-L1 Ab + cab + CD-k30		60	0	4	3	3	60	2	2	1	5	70	17	100
anti-CTLA-4 Ab + cab + CD-k30		90	0	1	1	8	90	0	1	0	9	100	0	100

Abbreviations: Ab, antibody; HDAC, histone deacetylase; ICI, immune checkpoint inhibitor; ORR, objective response rate; PD, partial disease; SD, stable disease; CR, complete response. * Relapse/recurrence was defined as tumor growth of at least 5-fold in mice with CR or PR response after the first tumor assessment. &: Second tumor assessment 10 days after the last drug administration. #: Mice resistant to CT26 challenge. Response evaluation criteria: fold-change of tumor size compared to baseline. PD: x ≥ 5, SD: 1 ≤ x < 5, PR: 0.5 ≤ x < 1, CR: x < 0.5.

**Table 3 ijms-23-10677-t003:** Male Balb/c mice (*n* = 129) bearing subcutaneous CT26 tumors treated with first-line anti-PD-1 therapy and anti-IgG (as negative control) antibody (2.5 mg/kg) once every 3 days for 3 doses.

Number of Mice	Response to First-Line Anti-PD-1 Antibody Therapy	Types of Drug Resistance to First-Line Anti-PD-1 Antibody Therapy
10	Treatment with anti-IgG antibody (as negative control)	N/A
17	Yes	Response *
91	NO	Primary resistance **
11	NO	HPD ***

Abbreviations: Ab, antibody; HPD, hyperprogressive disease; PD-1, programmed death-ligand 1; *: Response rate (CR% plus PR%): 17/119, 14.3%; ** Primary resistance rate: 91/119, 76.5%. *** HPD rate: 11/119, 9.2%.

**Table 4 ijms-23-10677-t004:** Response rates after treatment with different second-line regimens in CT26-bearing mice with primary resistance, acquired resistance, or HPD to first-line anti-PD-1 antibody treatment.

Therapeutic Resistance	Regimens	Initial Tumor Volume (mm^3^)	ORR (%)	PD	SD	PR	CR	ORR (%) ^&^	PD ^&^	SD ^&^	PR ^&^	CR ^&^	Survival Rate (%)	Relapse * (%)	Immunity # (%)
Primary resistance	anti-PD-1	396	0	4	1	0	0	0	5	0	0	0	0	-	-
	Anti-CTLA-4		0	0	7	0	0	0	2	5	0	0	0	-	-
	Anti-CTLA-4 + CC-02		37.5	1	4	0	3	37.5	2	3	0	3	37.5	0	100
	Anti-CTLA-4 + reg+ CD-k30		62.5	0	2	1	5	87.5	0	1	0	7	87.5	0	100
	Anti-CTLA-4 + cab + CD-k30		57.1	0	3	1	3	100	0	0	3	4	71.4	0	100
HPD	Anti-CTLA-4 + cab + CDHCl	669	18.1	0	9	0	2	45.4%	3	3	3	2	45.4	0	100

Abbreviations: Ab, antibody; HPD, hyperprogressive disease; PD-1, programmed death-ligand 1; ORR, objective response rate; PD, partial disease; SD, stable disease; CR, complete response. * Relapse/recurrence was defined as tumor growth of at least 5-fold in mice with CR or PR response after the first tumor assessment. &: Second tumor assessment 10 days after the last administration of second-line treatment. #: Mice resistant to CT26 challenge. -: Not tested Response evaluation criteria: fold-change of tumor size compared to baseline, PD: x ≥ 5, SD: 1 ≤ x < 5, PR: 0.5 ≤ x < 1, CR: x < 0.5.

## Data Availability

Data are available upon reasonable request.

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
