# Peer review of "Chidamide plus Tyrosine Kinase Inhibitor Remodel the Tumor Immune Microenvironment and Reduce Tumor Progression When Combined with Immune Checkpoint Inhibitor in Naïve and Anti-PD-1 Resistant CT26-Bearing Mice"

_ijms, 2022, doi:10.3390/ijms231810677_

Round 1

Reviewer 1 Report

The manuscript by Chen et al, investigate the effect of combination therapy using TKI, HDACi and ICI on ORR and OS in tumor bearing mouse. Authors have also detected TME remodeling through changes in transcriptional and translation regulation with special focus on modulation of immune populations like immunosuppressive cells, restoring T-cell activation T-cell activation. Authors have analyzed changes in immune regulatory genes by RNA-seq and GSEA. My specific comments are as follows.

 Major concerns:

Current study is involves detecting therapeutic combination of TKI, HDACi and ICI on mouse tumor beating CT26 tumor model. Including other tumor models such as lung and breast tumor mouse models to detect combinatorial effect of TKI, HDACi and ICI would provide insight for therapeutics targets for multiple cancer types.

According to Figure 2E, the anti-PD-1 antibody + regorafenib + chidamide-k30 triple combination increased CD8+ T cells infiltration in TME. What is phenotypic and functional characteristics or state of these CD8T cells? Authors should add data from cytokine assay e.g., IFN-y or TNF-a and data from Phenotypic characterization of CD8 T cells using multiple markers.

In Figure 2, Did authors looked at conventional and plasmacytoid Dendritic cell populations in tumor microenvironment while CT26 tumor bearing mice treated with by anti-PD-1 +TKI +CD-k30 treatment?

It will be interesting to detect the alteration in tumor infiltrating CD3, CD4, CD8, and Treg, CD11b, PMN-MDSC and M-MDSC cell population in tumor from CT26 tumor bearing mice treated with anti-CTLA4  or anti-PD-L1 combined with Cabozantinib or Regorafenib plus Chidamide-k30.

 What is the rationale behind doing RNA seq for gene expression on day 12 after starting second-line treatment from tumor samples treated with anti-CTLA-4 Ab combined with CC-02 or Regorafenib/Cabozantinib plus Chidamide-k30 in Figure 7 and 8?

In Figure 6, have author detected effect of aCTLA4 plus anti-PD-1 treatment on tumor of CT-26 tumor bearing mice? Data for tumor growth and OS/ORR should be added from treatment with aCTLA4 plus anti-PD-1 to make more clear comparison for different treatment regimen.

The data from rechallenge experiment shown in Table 4. However, tumor growth curves should be added from rechallenge experiment for much clear presentation of data.

Minor concerns:

High resolution image should be provided for all the figures in the manuscript. Additionally font size in the figure should be increased.

In the introduction section, line 91-93, Reference 9,17 doesn’t match with the statement, “Tumors associated with HPD showed, TIM3 overexpression in tumor-infiltrating CD8+ T cells, after PD-1 blockade treatment”. Correct citation should be added for association between TIM3 overexpression in tumor-infiltrating CD8+ T cells and PD-1 blockade.

In line 303, “As shown in S-Figure 1A–C, lenvatinib, cabozantinib, chidamide, regorafenib, axitinib, and above 4 drugs were combined with chidamide therapy”, should be corrected with, “lenvatinib, cabozantinib, regorafenib, axitinib, and above 4 drugs were combined with chidamide therapy”.

In Supplementary figure 2A-C, author should provide rational behind combining celecoxib, is a selective cyclooxygenase (COX)-2 inhibitor with ICI, TKI and HDACi in mouse tumor growth. More background should be added on this in introduction section.

 Full name of few words such as TIE2, CSF1R, RCC, HCC,mCC should be added.

Reviewer 2 Report

At present form, the readability of this manuscript is very low. Could the Authors improve the Figures’ quality? There are many of data, and analysis of the pictures is challenging. Thus, the Authors should re-built the manuscript to present their interesting data correctly. Only then will it be possible to evaluate the content of the article.

Author Response

At present form, the readability of this manuscript is very low. Could the Authors improve the Figures’ quality? There are many of data, and analysis of the pictures is challenging. Thus, the Authors should re-built the manuscript to present their interesting data correctly. Only then will it be possible to evaluate the content of the article.

Ans: thanks for your comment.

We have improved and re-builded the Figures, and revised content.

Reviewer 3 Report

Since I am biochemist and immunologist my review will not consider the bioinformatic part of the research.

General remarks : Performed research consider use of combined small molecules and immune cells with triple regimen comprising epigenetic immunomodulator  (chidamide: anti-angiogenesis class l histone deacetylase inhibitor)  with VEGF receptor tyrosine kinase inhibitor (cabozantinib/regorafenib) and immune checkpoint inhibitors  (anti-PD-1 and anti-CTLA-4 antibodies) for TME remodeling in cancer therapy . Investigations were done in vivo with murine colon carcinoma (CT26) allograft models (CT26 tumor- bearing mice) Manuscript is well written and despite the complex  experimental protocols obtained results are clearly presented and explained. 

Presented data supply our knowledge considering effect TME on cancer development and have also practical application in clinical studies .

Introduction:Introduction comprises all necessary informations concerning present knowledge about  immune cells of tumor environment, both, limfocyte  and myeloid lineage, immunesuppresive cancer activity, investigated inhibitors and immune therapy achievements with proper references.

Materials and Methods: All ethical conditions were fulfil and animals treatment and experimental  protocols very well described.  Also  necessary informations considering sources of applied reagents and antibodies and treatment diagram are supplied. All experiments were done with use of modern equipment and assay methods and result significance calculations performed with proper  programs.

Results: Results comprise the effect of triple regiment treatment on immune parameters of CT26 tumor- bearing mice including innate and adaptative immune response parameters as T cell-  mediated cytotoxicity, angiogenesis, chemokine activity, cytokine activity, leukocyte migration, and cell chemotaxis. Analysis of the numbers of chemokine and cytokine  genes  upregulated and downregulated after each regimen treatment  was also performed.

All Figures and Table are clearly  described

Fig.2. No data concerning the statistical significance marked as **

Presented results suggest  that regimen comprising anti-PD-1 antibody combined with multi-kinase inhibitor, such  as regorafenib or cabozantinib, plus chidamide-k30 activated a specific immune modulation of  the TME and could improve the tumor response rate, avoid recurrence, and induce  immune memory that provide rationale for clinical trias. Particularly important is observation that it may be an  important combination therapy to address resistance induced by first-line anti-PD-1  therapy.

Discussion: Discussion is too long due to repetitions of Introduction and Results  informations.But considering the complex subject of the investigations it may be accepted

Author Response

Comments and Suggestions for Authors

Since I am biochemist and immunologist my review will not consider the bioinformatic part of the research.

General remarks : Performed research consider use of combined small molecules and immune cells with triple regimen comprising epigenetic immunomodulator  (chidamide: anti-angiogenesis class l histone deacetylase inhibitor)  with VEGF receptor tyrosine kinase inhibitor (cabozantinib/regorafenib) and immune checkpoint inhibitors  (anti-PD-1 and anti-CTLA-4 antibodies) for TME remodeling in cancer therapy . Investigations were done in vivo with murine colon carcinoma (CT26) allograft models (CT26 tumor- bearing mice) Manuscript is well written and despite the complex  experimental protocols obtained results are clearly presented and explained. 

Presented data supply our knowledge considering effect TME on cancer development and have also practical application in clinical studies .

Introduction: Introduction comprises all necessary informations concerning present knowledge about  immune cells of tumor environment, both, limfocyte  and myeloid lineage, immunesuppresive cancer activity, investigated inhibitors and immune therapy achievements with proper references.

 Ans: thanks for your comment.

Materials and Methods: All ethical conditions were fulfil and animals treatment and experimental  protocols very well described.  Also  necessary informations considering sources of applied reagents and antibodies and treatment diagram are supplied. All experiments were done with use of modern equipment and assay methods and result significance calculations performed with proper  programs.

 Ans: thanks for your comment.

Results: Results comprise the effect of triple regiment treatment on immune parameters of CT26 tumor- bearing mice including innate and adaptative immune response parameters as T cell-  mediated cytotoxicity, angiogenesis, chemokine activity, cytokine activity, leukocyte migration, and cell chemotaxis. Analysis of the numbers of chemokine and cytokine  genes  upregulated and downregulated after each regimen treatment  was also performed.

All Figures and Table are clearly  described

Fig.2. No data concerning the statistical significance marked as **

 Ans: thanks for your comment.

We have corrected the issue by adding the definition in Figure 2 legend (line 1057-1059).

Presented results suggest  that regimen comprising anti-PD-1 antibody combined with multi-kinase inhibitor, such  as regorafenib or cabozantinib, plus chidamide-k30 activated a specific immune modulation of  the TME and could improve the tumor response rate, avoid recurrence, and induce  immune memory that provide rationale for clinical trias. Particularly important is observation that it may be an  important combination therapy to address resistance induced by first-line anti-PD-1  therapy.

Ans: thanks for your comment.

Discussion: Discussion is too long due to repetitions of Introduction and Results  informations. But considering the complex subject of the investigations it may be accepted

 Ans: thanks for your comment.

Round 2

Reviewer 1 Report

 In the revised version of manuscript, major and minor concerns have been addressed.

I have few minor comments:

Improved the quality of  tumor growth curves should be added in figure 1E, 4E and 5G ,figure axis legends font should be increased and symbol lablling for each indivisual grpwth curve shpuld be provided. 

Author Response

Thanks for your comment.

We have added the quality of tumor growth curves in figure 1E, 4E and 5G.

And also revised manuscript at line 1045-1050, 1093-1096, 1129-1132, 1139-1140